# Subclass-Dominant Label Noise: A Counterexample for the Success of Early Stopping

**Yingbin Bai**[1]    **Zhongyi Han**[2]    **Erkun Yang**[3]    **Jun Yu**[4]
**Bo Han**[5]    **Dadong Wang**[6]    **Tongliang Liu**[1] *

[1]Sydney AI Centre, University of Sydney; [2]Mohamed bin Zayed University of Artificial Intelligence;
[3]Xidian University; [4]University of Science and Technology of China;
[5]Hong Kong Baptist University; [6]CSIRO

## Abstract

In this paper, we empirically investigate a previously overlooked and widespread type of label noise, subclass-dominant label noise (SDN). Our findings reveal that, during the early stages of training, deep neural networks can rapidly memorize mislabeled examples in SDN. This phenomenon poses challenges in effectively selecting confident examples using conventional early stopping techniques. To address this issue, we delve into the properties of SDN and observe that long-trained representations are superior at capturing the high-level semantics of mislabeled examples, leading to a clustering effect where similar examples are grouped together. Based on this observation, we propose a novel method called NoiseCluster that leverages the geometric structures of long-trained representations to identify and correct SDN. Our experiments demonstrate that NoiseCluster outperforms state-of-the-art baselines on both synthetic and real-world datasets, highlighting the importance of addressing SDN in learning with noisy labels. The code is available at `https://github.com/tmllab/2023_NeurIPS_SDN`.

## 1 Introduction

The popularity of giant neural networks in recent years has emphasized the importance of vast amounts of annotated data [42, 11, 41]. However, manually annotating large amounts of data can be prohibitively expensive, leading researchers to explore alternative methods for collecting data such as web crawling [57] and crowdsourcing [45] that can be more cost-effective. Unfortunately, these methods may produce inaccurate data, particularly when noisy labels are present, thereby undermining the performance of deep neural networks (DNNs). As a result, the research community has increasingly focused on developing approaches to mitigate the impact of noisy labels and prevent the degeneration of DNNs [16, 25, 13].

Most existing approaches for dealing with noisy labels broadly fall into two categories: noise-modeling-based methods and memorization-effects-based methods. Noise-modeling-based methods [51, 53, 32, 47, 55] usually construct a transition matrix to exploit connections between given noisy labels and underlying clean labels. While these methods offer strong theoretical guarantees that the corrected risk closely approximates the risk associated with training on clean data, accurate estimation of the transition matrix relies on numerous assumptions that are difficult to satisfy in real-world scenarios [40, 56]. Memorization-effects-based methods [17, 43] mainly employ early stopping to identify confident examples based on the phenomenon that DNNs will fit the easy examples (i.e., the majority of examples with similar characteristics) first and then memorize the complex examples (i.e., the minority of examples with diverse characteristics) [58, 1, 22]. Recently, some of them [30, 34, 3], combined with semi-supervised learning techniques [5, 23], have achieved state-of-the-art results.

---

*Correspondence to Tongliang Liu (tongliang.liu@sydney.edu.au)

37th Conference on Neural Information Processing Systems (NeurIPS 2023).

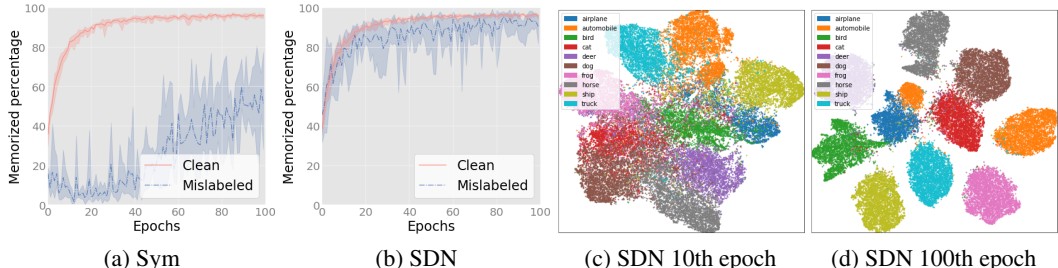

|          |          |          |          |
|:--------:|:--------:|:--------:|:--------:|
| (a) Sym  | (b) SDN  | (c) SDN 10th epoch | (d) SDN 100th epoch |

Figure 1: **(a, b)** It is evident that early stopping can effectively separate symmetric label noise (Sym), but it proves ineffective for subclass-dominant label noise (SDN) as the network initially memorizes mislabeled examples.**(c, d)** Using the same settings as in (b), it becomes apparent that representations from the 100th epoch exhibit superior geometric structures compared to those from the 10th epoch. Specifically, the boundaries between classes become clearer, allowing for better distinguishability of correctly labeled and mislabeled examples.

In this paper, we present a new type of label noise, subclass-dominant label noise (SDN), which is overlooked by existing studies. SDN refers to label noise in which mislabeled examples dominate at least one subclass, whose examples exhibit distinct characteristics compared to others in the same class. Real-world datasets are prone to experiencing SDN due to several factors. The likelihood of human mislabeling is influenced by various object characteristics such as shape, context, and part [50, 33, 7, 44, 6]. Moreover, classification tasks usually define classes based on typical or common examples. However, in reality, numerous special sub-classes can exist, sharing common characteristics with other classes. For instance, while whiskers typically play a vital role in distinguishing between cats and dogs, Sphynx cats, unlike other cat breeds, do not possess whiskers. An inexperienced annotator may mislabel the majority of Sphynx cats as dogs, leading to SDN in the dog class.

To evaluate the detrimental effects of SDN, we construct an SDN dataset and conduct an experiment on it. Specifically, we subdivide the aircraft class in CIFAR-10 [27] into two sub-classes based on their status: landed and flying. We generate the SDN dataset by flipping the labels of all landed airplane examples to the automobile class, leading to an overall noise rate of 2.7%. Our experiment involves training a ResNet-18 model [19] for 100 epochs using cross-entropy loss and extracting confident examples at each epoch. We identify confident examples based on the criteria described in [43]. To quantify the extent to which the network has memorized correctly labeled and mislabeled data, we compute the ratio of confident examples that include mislabeled data to the total number of mislabeled examples in the training dataset. As shown in Figure 1(a, b), early stopping fails to identify mislabeled examples because the network has already memorized the mislabeled examples in SDN. This outcome can be attributed to the fact that DNNs excel at learning from the majority of examples that share similar characteristics first. When the majority of examples in a clean subclass are mislabeled, these mislabeled examples often dominate the gradient and are learned first by DNNs, thereby posing challenges for methods that rely on early stopping.

To overcome this challenge, we explore a novel method named NoiseCluster, which is capable of distinguishing correctly labeled and mislabeled examples in SDN. The fundamental principle of NoiseCluster is rooted in the concept of later stopping. Contrary to the conventional belief that noisy data increasingly degrade representations over time, our findings suggest that the long-trained representations obtained from later stopping are more effective at capturing the high-level semantics of noisy examples. This leads to a clustering effect where the embeddings of noisy examples with similar characteristics are drawn closer together, as shown in Figure 1(c, d). Motivated by this observation, NoiseCluster utilizes the geometric structures of long-trained representations to identify and correct mislabeled examples of SDN. Specifically, we first extract features from the penultimate layer of the network after halting the training progress with later stopping. We then group these features by class, based on feature density, and identify the largest group as clean due to its high likelihood of being correctly labeled, while treating the remaining groups as potentially mislabeled. To determine the label of a potentially mislabeled group, we assess its similarity to surrounding examples according to set distance and assign it the class with the most similar examples.

The key contributions are summarized as follows:

- We introduce subclass-dominant label noise (SDN) for the first time, highlighting its prevalence in real-world datasets and its potential to render early stopping ineffective.

- We propose a novel method, NoiseCluster, which leverages the geometric structures of long-trained representations to identify and correct mislabeled examples in SDN.

- Empirical results demonstrate the effectiveness of NoiseCluster on both synthetic and real-world datasets, establishing it as a solid baseline for future studies.

## 2 Related Work

Learning with noisy labels is a vibrant research area aimed at improving model robustness. Two main avenues of research have emerged: noise-modeling-based and memorization-effects-based methods.

**Noise-modeling-based methods.** From a noise modeling perspective, label noise can be categorized into three types: random classification noise (RCN), class-conditional label noise (CCN), and instance-dependent label noise (IDN). RCN maintains a static flip rate for binary classification [35]. CCN assumes that the noise generation is solely dependent on classes and independent of instances [40, 56, 48, 9]. The generation of IDN depends on both classes and instances, making IDN the most general type of label noise [50, 10, 62, 36]. The common philosophy behind noise modeling is to estimate the noise transition matrix. Xia et al. [50] assume annotators may make mistakes on the parts of instances rather than on whole instances, and replace the estimation of the instance-dependent transition matrix with the transition matrices for parts; Yang et al. [55] propose a method to generate Bayes optimal labels and estimate the transition matrix between Bayes optimal labels and noisy labels. Although noise-modeling-based methods do not theoretically require early stopping, many of them implicitly employ it to accurately estimate the transition matrix [40, 51, 53, 56, 55]. As a result, many existing methods struggle to handle SDN.

**Memorization-effects-based methods.** Arpit et al. [1] find that Deep Neural Networks (DNNs) tend to learn simple (clean) examples first, then gradually memorize complex (noisy) examples. This observed phenomenon has inspired a subset of methodologies grounded in understanding and leveraging memorization effects [17, 2, 38, 22, 24]. JointOptim [43] introduces a pseudo-labeling technique known as hard-label, which regards the model predictions consistent with noisy labels as confident examples and uses them to train the model. DivideMix [30] utilizes Gaussian Mixture Model (GMM) to identify confident examples as labeled data and treat the remaining examples as unlabeled data for semi-supervised learning (SSL). Furthermore, some methods focus on the latent representations extracted from previous layers, as the final layer is particularly vulnerable to the influence of noisy examples [29, 46, 26, 3]. TopoFilter [46] utilizes k-NN to explore the topological information of the latent representations and then selects the largest component as clean data. Kim et al. [26] propose a noise detection method called FINE that utilizes Eigen decomposition and the principal component to select clean examples. PES [3] takes advantage of the fact that the effects of noisy examples tend to decrease as the layer number increases and improves early stopping by re-learning later layers based on the latent representations. NoiseCluster also exploits the geometric structures of the latent representations. However, unlike previous methods that rely on early stopping, NoiseCluster takes advantage of later stopping to extract long-trained representations, enabling the identification and correction of mislabeled examples in SDN.

## 3 SDN Analysis

In this section, we begin by defining subclass-dominant label noise (SDN). Next, we introduce a specially crafted dataset, CIFAR20-SDN, for exploring SDN, and compare it with two established real-world datasets. Finally, we delve into the characteristics of SDN from two distinct perspectives: its adverse effects on early stopping and its negative impact across various layers.

**Definition of SDN.** Let $\widetilde{D} = \{(x_i, \widetilde{y}_i)\}_{i=1}^n$ be the noisy dataset, where $x_i \in X$ denotes input data, $\widetilde{y}_i \in \{1, ..., C\}$ denote a noisy label, and $C$ is the total number of classes. $x_i$ is associated with a latent (not provided) clean subclass label $b_i \in \{1, ..., B\}$, and $B$ is the total number of sub-classes in this dataset. It is assumed that each clean subclass $b$ corresponds to a single clean class $c$. Subclass-dominant noise (SDN) refers to a type of label noise in which mislabeled examples dominate at least one subclass. Specifically, within a clean subclass $b$, examples could be flipped into different classes,

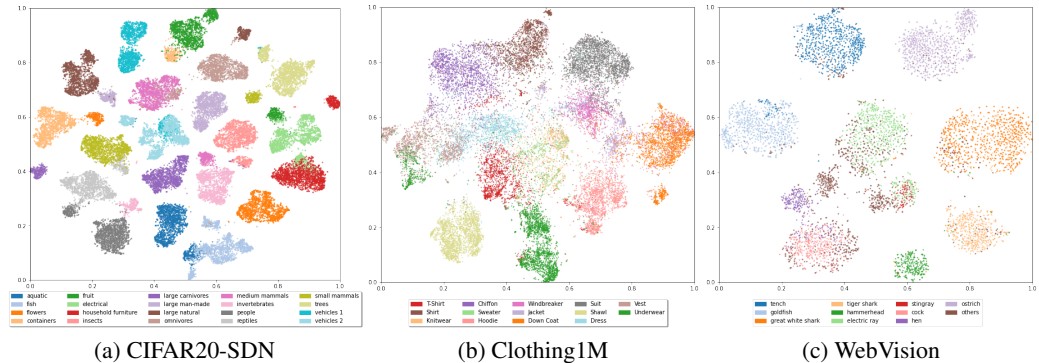

|  | (a) CIFAR20-SDN | (b) Clothing1M | (c) WebVision |
|---|---|---|---|

Figure 2: Comparison of feature distribution between CIFAR20-SDN and two real-world noisy datasets, with colors corresponding to clean labels. For Clothing1M, we utilize a provided evaluation set containing both clean and noisy labels for each example, employing the early stopping process described in Section 5.1. For WebVision, a DivideMix model is employed to generate features for the first ten classes; "clean" labels for these are inferred using models pretrained on ImageNet.

Table 1: Comparison of noisy impacts on different layers with different types of label noise. Training with clean labels terms as Clean; "Sym" is short for symmetric label noise; "IDN" is short for instance-dependant label noise; and "Pair" is short for pairflip label noise. "-XX" indicates the noise rate. The mean and standard deviation (percentage) computed over five runs are presented.

| NOISE TYPE | CLEAN | SYM-20 | IDN-20 | PAIR-20 | SDN-20 |
|---|---|---|---|---|---|
| FINAL | 85.17±0.32 | 72.25±0.21 | 72.41±0.22 | 73.39±0.48 | 68.50±0.19 |
| PENULTIMATE | 85.28±0.35 | 73.43±0.67 | 76.51±0.89 | 78.77±0.34 | 81.67±0.17 |

including its own clean class. If the class $j \in \{1, ..., C\}$ with the most flipped examples does not equal to the clean class $c$, then the subclass $b$ is considered to be dominated by the examples flipped into noisy class $j$.

**CIFAR20-SDN.** To facilitate research on SDN, we introduce CIFAR20-SDN, a representative SDN dataset built from CIFAR-100, which provides 20 class labels and 100 subclass labels. The generation of CIFAR20-SDN is similar to that of Pairflip label noise [17]. Specifically, for each class, a certain percentage of labels in the last subclass are flipped to the subsequent class. For example, the first class in CIFAR20 is termed "aquatic mammals", which comprises five sub-classes: "beaver", "dolphin", "otter", "seal", and "whale". Labels from the "whale" subclass are randomly flipped to the subsequent "fish" class. As a result, any "whale" labeled as "fish" is deemed an instance of SDN. Note that only the class labels are utilized for model training and evaluation.

To assess the similarity between CIFAR20-SDN and real-world noisy datasets, we compare the feature distribution of CIFAR20-SDN with two other real-world noisy datasets, Clothing1M [52] and WebVision [31]. Specifically, we directly visualize the features of CIFAR20-SDN, as it exclusively contains SDN, while for the two real-world datasets, we visualize the features of confident examples extracted using early stopping methods. As shown in Figure 2, it is clear that CIFAR20-SDN successfully emulates the properties of the two real-world datasets, with mislabeled examples clustering together and maintaining a distance from their noisy classes.

**Negative effects of SDN on different layers.** To thoroughly study the negative impacts of SDN on various layers, we train a PreAct ResNet-18 model on CIFAR20-SDN with four types of label noise. We report the highest test accuracy achieved during training as an indicator of the quality of the representations from the final layer. Additionally, to assess the representation quality from the penultimate layer, we freeze the other layers, re-initialize and retrain the final layer using clean labels.

Table 1 reveals that SDN significantly impacts the final layer more detrimentally than other types of label noise, resulting in the lowest test accuracy, which is roughly 80% of the accuracy achieved on the clean dataset. Conversely, when considering the penultimate layer, the outcomes achieved with SDN surpass those obtained with any other type of label noise, indicating that previous layers are less adversely impacted even after 300 epochs of training. These findings underscore the importance of utilizing the representations extracted from the penultimate layer in addressing SDN.

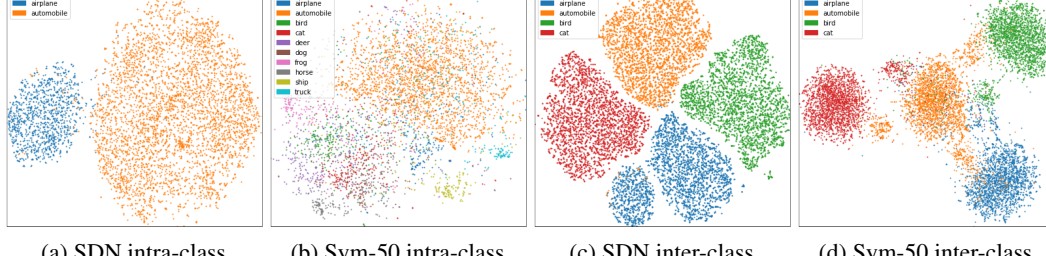

| (a) SDN intra-class | (b) Sym-50 intra-class | (c) SDN inter-class | (d) Sym-50 inter-class |

Figure 3: (a, b) illustrates the intra-class relationships between correctly labeled and mislabeled examples in a noisy class, while (c, d) demonstrates the inter-class relationships between mislabeled examples and their corresponding latent clean classes. To better visualize the intra-class feature relationships, we use the features from a noisy *automobile* class and visualize them with clean labels. Similarly, to effectively visualize inter-class feature relationships, we use features from four classes, as 2D images cannot adequately represent high-dimensional relationships.

**Advantages of long-trained representations**. We extend the experiments depicted in Figure 1 and visualize the features extracted from the networks trained for 200 epochs. As shown in Figure 3(a, b), features within a noisy class are clustered based on their semantics, which suggests that networks can learn the semantics of mislabeled examples even when trained with noisy labels. Moreover, Figure 3(c, d) shows that features of mislabeled examples tend to be close to the features with corresponding clean classes, implying that the geometric structures of the features can be utilized to identify their clean classes. Finally, a comparison of the four sub-figures reveals that the features extracted from SDN exhibit superior geometric structures compared to those extracted from symmetric label noise. This observation suggests that the properties of long-trained representations are particularly advantageous when dealing with SDN.

## 4 Methodology

To overcome the challenges posed by SDN, we introduce a novel method called NoiseCluster, which leverages the geometric structures of long-trained representations. NoiseCluster consists of two main steps: initially, it identifies potentially mislabeled examples by clustering similar instances into multiple sets. Subsequently, it performs label correction by computing set distance and reassigning the labels of potentially mislabeled examples.

### 4.1 Identify SDN

NoiseCluster begins with training on $\widetilde{D}(X, \widetilde{Y})$ and then halts the training with later stopping, whose stopping point is much later than that of early stopping, allowing for thorough exploitation of knowledge in noisy data. We then extract the latent feature $Z$ of the input data $X$ from the penultimate layer and reduce its dimensionality to two dimensions using t-SNE [39], resulting in a new feature denoted as $\hat{Z}$. To separate mislabeled data, we explore the geometric structures of $\hat{Z}_c$, where $c \in \{1, ..., C\}$ and $C$ is the total number of classes. To handle the unknown number of clusters in each class, we employ DBSCAN [12], a feature density-based clustering algorithm, to cluster the feature $\hat{Z}_c$ into $K$ groups $U_K^c$. The largest group, which is most likely to contain correctly labeled examples, is identified as the clean group, while the remaining groups are regarded as potentially mislabeled groups and denoted as $U_{K-1}^c$.

**Stopping point selection**. Although NoiseCluster can be applied to a fully trained model, we find that further training on the original noisy data does not constantly improve the representations after the training becomes stable. Therefore, to reduce training time, we incorporate our method into the middle stage of the training process.

### 4.2 Label Correction

Simply removing all examples in $U_{K-1}^c$ can result in the loss of many important examples. To address this challenge, we treat examples in a potentially mislabeled group $U_k^c$ as a cohesive unit, where all

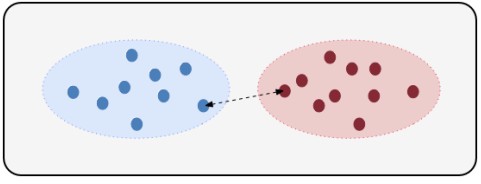 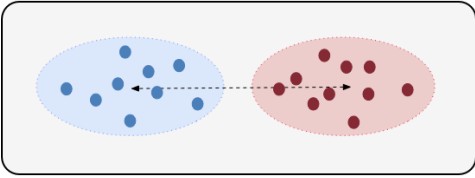

| (a) Set distance | (b) Mean distance |

Figure 4: The figure compares two methods for quantifying the similarity between two sets. Compared to mean distance shown in (b), set distance can more accurately measure the similarity between a potentially mislabeled group and a subset of similar examples within the class sets, particularly when the classes have high diversity.

examples share the same label. We then assess the similarity between this group and the surrounding examples by computing distances and assigning examples in the group to the class with the most similar examples. Specifically, due to the clustering effects of long-trained representations, we treat the examples in $U_k^c$ as a set, and form multiple class sets $O_b$ from the surrounding examples, where $b \in \{1, ..., C\}$. We measure the distance between $U_k^c$ and $O_b$ using set distance, which is defined as the minimum Euclidean distance between any feature $z_q, z_p$ from sets $U_k^c$ and $O_b$, as depicted in Figure 4. The formula for set distance is defined as follows:

$$d(U_k^c, O_b) = \inf_{p \in U_k^c, q \in O_b} \{\|z_q - z_p\|_2\},\tag{1}$$

where $\|\cdot\|_2$ indicates $L2$ Norm. In practice, to mitigate the potential impacts of mislabeled examples close to the set $U_k^c$, we utilize a specified number of closest distances, denoted as $ClosePoint$, and compute their average. We then correct the labels in $U_k^c$ with the category of the class set with the closest distance, $d_{kb}$, which is defined as follows:

$$d_{kb} = \inf_{b \in \{1, ..., C\}} \{d(U_k^c, O_b)\}.\tag{2}$$

After acquiring the corrected data, we use them to continue training the network for the rest of the epochs. The whole process of NoiseCluster is summarized in Algorithm 1.

**Combine with SSL**. Semi-supervised learning (SSL) has enabled numerous methods to attain impressive benchmarks in LNL [30, 34, 26]. NoiseCluster can also take advantage of SSL techniques to enhance its performance. Specifically, for each class $c$, DBSCAN divides features $\hat{Z}_c$ into $K$ distinct groups $U_K^c$ and assigns a group index $g_i^c$ to each feature $\hat{z}_i^c$. Examples that cannot be clustered into any group are regarded as "noise". To integrate SSL, we treat the DBSCAN noise as the unlabeled data and the remaining examples as the labeled data. Accordingly, we can summarize the labeled and unlabeled data sets as follows:

$$\begin{cases} \mathcal{D}_l = \{(x_i, \widetilde{y}_i) \mid g_i^c = 1, ..., K_c\} \\ \mathcal{D}_u = \{x_i \mid g_i^c \neq 1, ..., K_c\} \end{cases}.\tag{3}$$

**Combine with early stopping**. Although NoiseCluster is primarily designed to address SDN, it can be seamlessly integrated with early stopping. To tackle a noisy dataset containing multiple types of label noise, we propose a two-stage training process. In the first stage, we employ early stopping to select confident data, thereby filtering out label noise that is sensitive to early stopping. In the second stage, we apply NoiseCluster to the confident data to identify and correct mislabeled examples in SDN. This two-stage strategy allows us to effectively address the challenges posed by a mixture of various types of label noise, resulting in enhanced performance on noisy datasets.

## 5 Experiments

### 5.1 Experimental Setting

**Datasets**. We evaluate the performance of NoiseCluster on a synthetic dataset, CIFAR20-SDN, and a real-world dataset, Clothing-1M [52]. To ensure a fair comparison with existing methods, we follow the settings of the previous methods [30, 34, 3] and conduct two sets of experiments on the synthetic

**Algorithm 1:** *NoiseCluster*

---

**Input**: Network $f_\theta$; Final layer $f_\xi$; Noisy training dataset $\widetilde{D}(X, \widetilde{Y})$; Number of epochs for long-trained $N$; Class number $C$; DBSCAN $Eps$ and $MinPts$.

**for** $i = 1, \ldots, N$ **do**
    Train $f_\theta$ and $f_\xi$ on $\widetilde{D}(X, \widetilde{Y})$;              `// standard training`

$\hat{Z} \leftarrow tSNE(f_\theta(X))$;
**for** $c \leftarrow 1$ **to** $C$ **do**
    $U_K^c \leftarrow DBSCAN(\hat{Z}_c, Eps, MinPts)$;          `// identify SDN`
    **for** $U_k^c$ **in** $U_K^c$ **do**
        **if** $U_k^c \neq largest$ **then**
            Compute set distance with Eq. (1);
            Update $\widetilde{Y}$ in $U_k^c$ with Eq. (2) ;
    $\mathcal{D}_l \leftarrow \mathcal{D}_l \cup U_k^c$;

Continually train $f_\theta$ and $f_\xi$ on $\mathcal{D}_l$ for the rest of the epochs.

---

Table 2: Comparison with state-of-the-art methods without SSL on CIFAR20-SDN. "-XX" indicates the noise rate. The mean and standard deviation computed over five runs are presented.

| METHODS | SDN-12 | SDN-16 | SDN-18 | SDN-20 |
|---|---|---|---|---|
| CE [19] | 70.42±0.32 | 67.38±1.31 | 66.28±0.51 | 65.23±0.60 |
| JOINTOPTIM [43] | 71.48±0.33 | 67.69±0.37 | 66.51±0.28 | 65.45±0.20 |
| DMI [53] | 70.67±0.64 | 66.95±0.61 | 65.86±0.32 | 64.35±0.52 |
| TOPOFILTER [46] | 69.04±0.47 | 64.78±0.49 | 63.52±0.36 | 62.97±0.62 |
| PTD-R-V [50] | 70.85±0.44 | 66.45±0.40 | 65.29±0.47 | 64.41±0.20 |
| VOLMINNET [32] | 72.45±0.38 | 67.78±0.43 | 66.20±0.24 | 65.20±0.41 |
| CDR [49] | 72.61±0.36 | 68.87± 0.16 | 67.79±0.28 | 67.00±0.25 |
| BLTM-V [55] | 70.46±0.47 | 66.57±0.50 | 64.76±0.42 | 63.96±0.46 |
| NOISECLUSTER | **80.20±0.80** | **78.77±1.01** | **77.01±0.34** | **72.67±0.68** |

dataset: one with semi-supervised learning (SSL) and one without SSL. In experiments without SSL, we reserve 10% of the training data as the validation set, while we utilize the entire training data for experiments with SSL. Clothing-1M contains over one million images gathered from the Internet. Additional details on implementation can be found in Appendix B.

**Network structure and optimization**. For CIFAR20-SDN, we employ ResNet-34 [19] for experiments without SSL and PreAct ResNet-18 [20] for experiments with SSL. During optimization, we train the model for 300 epochs, using a learning rate of $2 \times 10^{-2}$, a single cycle of cosine annealing [37], a momentum of 0.9, and a weight decay of $5 \times 10^{-4}$. We utilize a batch size of 128 and a stopping epoch of 80, with a ClosePoint value of 20. For DBSCAN hyperparameters, Eps and MinPts are set to 0.02 and 100, respectively.

For Clothing1M, we employ a ResNet-50 pre-trained on ImageNet[28], and train the network for only one epoch. Specifically, we first utilize 95% of the training data to train the network. Then, we use early stopping to select confident data from the remaining 5% of the training data and apply NoisyCluster to this data. For optimization, we use SGD with a learning rate of 0.01, a momentum of 0.9, a weight decay of $10^{-4}$, and a batch size of 64. ClosePoint, Eps, and MinPts are set to 20, 0.04, and 100, respectively.

## 5.2 Comparison with State-of-the-art Methods

**Approaches without SSL.** To begin our comparison, we evaluate our method against state-of-the-art (SOTA) approaches that do not employ SSL. As shown in Table 2, it becomes evident that many of the methods struggle with SDN, with some even underperforming in comparison to standard training with Cross-Entropy (CE) loss. CDR [49] performs better than its counterparts, but it requires a known noise rate, which remains challenging to accurately estimate in SDN. Our method outperforms all baselines, delivering superior performance. Specifically, our proposed method surpasses CDR by 5.67% in classification accuracy when a noise rate is 20%, and the margin further extends to

Table 3: Comparison with state-of-the-art methods with SSL on CIFAR20-SDN. "-XX" indicates the noise rate. The mean and standard deviation computed over five runs are presented.

| METHODS | SDN-12 | SDN-16 | SDN-18 | SDN-20 |
|---|---|---|---|---|
| CE [20] | 74.27±0.40 | 70.53±0.38 | 69.27±0.12 | 68.50±0.19 |
| MIXUP [59] | 75.83±0.41 | 71.65±0.17 | 70.29±0.24 | 69.37±0.17 |
| ELR+ [34] | 75.22±0.37 | 71.09±0.30 | 70.18±0.23 | 69.55±0.15 |
| DIVIDEMIX [30] | 79.06±0.26 | 72.80±0.30 | 70.66±0.51 | 70.27±0.11 |
| F-DIVIDEMIX [26] | 79.71±0.32 | 74.00±0.83 | 71.69±0.22 | 69.59±0.43 |
| PES SEMI [3] | 80.27±0.30 | 73.95±0.24 | 72.78±0.22 | 71.72±0.45 |
| NOISECLUSTER+ | **82.20±0.93** | **80.35±1.23** | **77.59±0.86** | **73.54±0.72** |

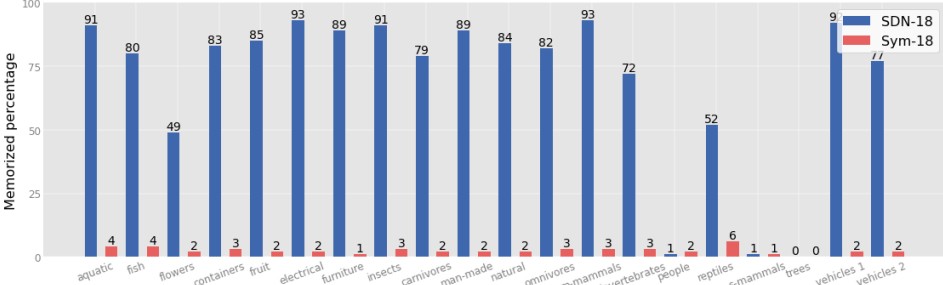

Figure 5: Evaluation memorized percentage per class obtained with DivideMix on CIFAR20-SDN (SDN) and symmetric label noise (Sym). "-XX" indicates the noise rate.

Table 4: Classification accuracy (%) on Clothing1M. Baseline results are taken from the original papers. * indicates the results obtained with an ensemble model.

| CE | DMI | JOINTOPTIM | TOPOFILER | PTD-R-V | VOLMINNET |
|---|---|---|---|---|---|
| 69.21 | 71.67 | 72.16 | 74.10 | 72.46 | 72.42 |
| BLTM-V | F-DIVIDEMIX* | PES | DIVIDEMIX* | ELR+* | NOISECLUSTER |
| 73.39 | 74.37 | 74.64 | 74.76 | 74.81 | **75.51** |

7.59% for a noise rate of 12%. Furthermore, in comparison with noise-modeling-based methods, our approach outperforms the state-of-the-art VolMinNet by over 7% across four noise rates.

**Approaches with SSL.** We integrate our method with semi-supervised learning (SSL) techniques, denoted as NoiseCluster+ and compare it against SOTA approaches that also leverage SSL. To ensure a fair comparison, we include the results of MixUp [59], which is used by all baselines to boost performance. The results are summarized in Table 3. Interestingly, our observations indicate that SSL contributes marginally to performance enhancement on CIFAR20-SDN. Notably, DivideMix [30] shows only a modest advancement over MixUp. In contrast, methods based on representations, such as F-DivideMix [26] and PES semi [3], deliver better outcomes by avoiding the use of the final layer, which is prone to overfitting to SDN. Conclusively, our method, NoiseCluster+, consistently outperforms all baselines by a substantial margin, underscoring that the proposed techniques are the primary factors leading to the improvements.

To better understand why current SOTA methods face difficulties when dealing with SDN, we conduct experiments with DivideMix and compute the ratio of confident examples that include mislabeled data to the total number of mislabeled examples in the training dataset at the end of training. As shown in Figure 5, DivideMix performs well on merely three classes. For the remaining classes, we can see that the confident data contains a substantial number of mislabeled examples, confirming that early stopping and other techniques utilized in DivideMix cannot effectively address SDN issues.

**Real-world dataset.** We perform experiments on a real-world dataset, Clothing1M, which is widely used as a benchmark due to its challenges. Many existing methods, though effective on noisy synthetic datasets, exhibit limited impact on Clothing1M. Our investigation suggests that this issue stems

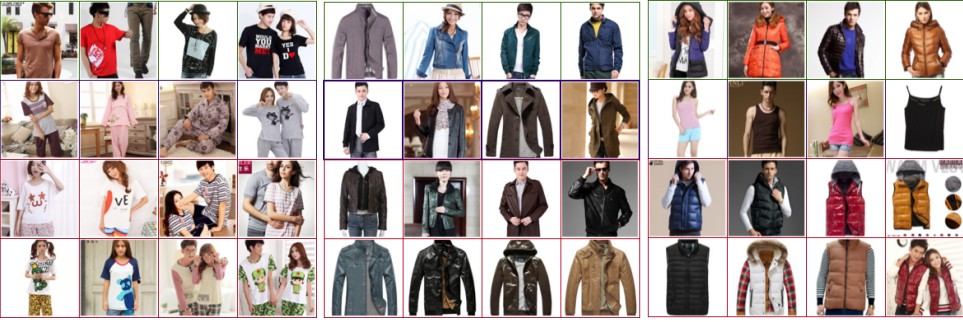

(a) Sleep T-Shirt(X:0.45, Y:0.1)    (b) Leather Jacket(X:0.75, Y:0.6)    (c) Down Vest(X:0.9, Y:0.3)

Figure 6: SDN in Clothing1M. (a) The first row displays images from the "T-Shirt" class. The second row shows images from the "Underwear" class. The third and fourth rows present images that are mislabeled as "Underwear". (b) The row order is as follows: "Jacket", "Windbreaker" (correct name:"trench coat"), "Leather Jacket". (c) The row order is as follows: "Down coat", "Vest" (correct name:"singlet" or "tank top"), "Down Vest". The coordinate values correspond to Figure 2 (b). It is worth noting that some label names within Clothing1M are inaccurate due to translation issues. After comparing images in the test set and Chinese label names, we put the correct names in parentheses.

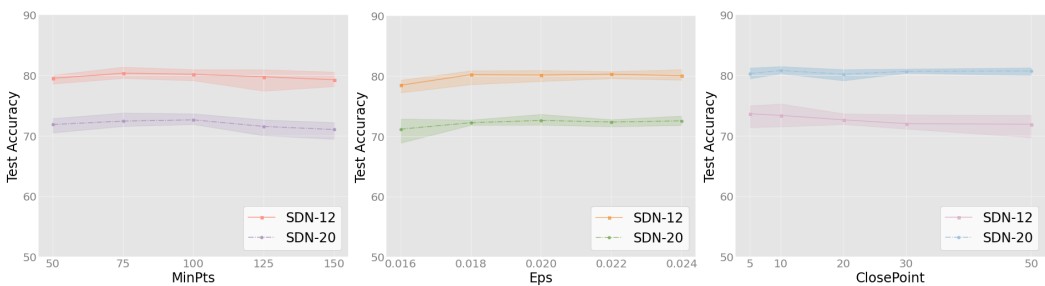

Figure 7: Ablation analyses of MinPts, Eps, and ClosePoint.

from the presence of SDN in Clothing1M, leading to suboptimal performance on the test data. As illustrated in Table 4, NoiseCluster significantly outperforms SOTA methods, achieving an accuracy of 75.51% without resorting to self/semi-supervised learning or ensemble network techniques.

To delve into the existence and underlying causes of SDN in real-world datasets, we manually generated t-SNE images, strictly adhering to the experimental process outlined in Section 5.1. We then showcase examples within clusters that are predominantly mislabeled. As Figure 6 illustrates, these mislabeled examples clearly belong to specific sub-classes, yet they share common characteristics with their associated noisy classes. These observations confirm the pervasive presence of SDN in real-world datasets. For additional details and further results, please refer to Appendix D.1.

### 5.3 Ablation Study

**Sensitivity of hyper-parameters.** We perform ablation studies on CIFAR20-SDN with noise rates of 12% and 20%. NoiseCluster employs DBSCAN [12], which involves two hyperparameters, Eps and MinPts. We find that the selection of these two hyperparameters is relatively straightforward when applied to feature representations due to the substantial margin between classes. Additionally, to reduce the negative impacts of mislabeled examples, we introduce a hyperparameter named ClosePoint, which controls the number of points used to compute the set distance. The results shown in Figure 7 indicate that ClosePoint, along with Eps and MinPts, has a negligible impact on the final outcomes, indicating robustness against hyperparameter changes.

**Self-supervised pretrained model.** As self-supervised Learning methods continue to mature [18, 15, 61, 4], the utilization of pretrained models from this approach has emerged as a promising strategy for addressing noisy labels, particularly since they are unaffected by such labels during the pre-training

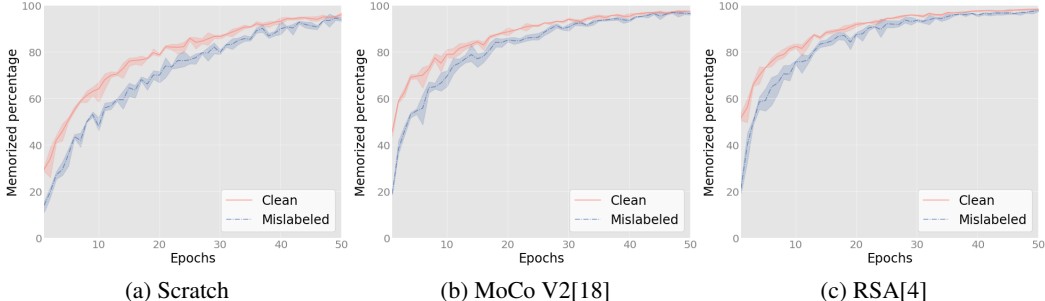

| (a) Scratch | (b) MoCo V2[18] | (c) RSA[4] |

Figure 8: Examining memorization trends when training from scratch compared to training with self-supervised pretrained models on CIFAR20-SDN with an 18% noise rate. These models were pretrained with MoCo V2 and RSA on the same dataset for 200 epochs, resulting in linear classification accuracies of $63.02 \pm 0.31$ and $77.67 \pm 0.27$, respectively. However, as the performance of pretrained models increases, the speed of overfitting to mislabeled data also becomes faster. The mean and standard deviation (percentage) computed over three runs are presented.

Table 5: Comparison with/without correcting labels on CIFAR20-SDN. "-XX" indicates the noise rate. The mean and standard deviation (percentage) computed over five runs are presented.

| METHODS | SDN-12 | SDN-20 |
|---|---|---|
| OURS | **80.20±0.80** | **72.67±0.68** |
| OURS W/O CORRECTION | 78.68±0.61 | 71.42±0.54 |

stage [14, 54, 21, 60]. This raises the pivotal question: Does employing an self-supervised pretrained model simplify the addressing of SDN? While a thorough investigation will be reserved for future research work, the succinct answer is no. As shown in Figure 8, self-supervised pretrained models offer initial advantages by facilitating rapid learning of clean examples. However, this rapid learning also leads to the quick memorization of mislabeled examples. Furthermore, after only training for several epochs, the gaps between the memorization rates for clean and mislabeled examples become much smaller than those observed when training from scratch, suggesting that early stopping with self-supervised pretrained models is less effective. We hope that our research will inspire a re-thinking of utilizing self-supervised pretrained models in noisy datasets.

**Importance of label correction.** To mitigate the bias introduced by learning with SDN, we employ a label correction technique. Given that DNNs have already memorized many mislabeled examples during the early learning phase, label correction becomes indispensable. In Table 5, we present a comparison of results with and without applying label correction and observe a performance improvement of over 1% for both low and high noise rates. These findings emphasize the importance of the label correction technique in addressing SDN.

## 6 Conclusion

In this paper, we empirically examine subclass-dominant label noise (SDN), a new type of label noise that commonly occurs in real-world datasets. SDN presents a significant challenge to conventional methods that rely on early stopping. To overcome this issue, we reveal that long-trained representations are more effective at capturing the semantic relationships between noisy examples. Building on this insight, we propose a novel approach, NoiseCluster, to identify and correct mislabeled examples in SDN. We hope our work will inspire the research community and pave the way for addressing label noise in real-world applications.

The primary limitation is that our proposed method is effective in addressing SDN cases where mislabeled examples have notably different semantics from correctly labeled examples. However, it could struggle in situations where correctly labeled and mislabeled examples exhibit close semantics, especially without any prior knowledge. We leave it as an open research topic for future investigation.

## Acknowledgments and Disclosure of Funding

The authors would like to thank the anonymous reviewers and the meta-reviewer for their constructive feedback and encouraging comments on this work. Yingbin Bai was supported by CSIRO Data61. Erkun Yang was supported in part by the National Natural Science Foundation of China under Grant 62202365, Guangdong Basic and Applied Basic Research Foundation (2021A1515110026), and Natural Science Basic Research Program of Shaanxi (Program No.2022JQ-608). Jun Yu was supported by the Natural Science Foundation of China (62276242), National Aviation Science Foundation (2022Z071078001), CAAI-Huawei MindSpore Open Fund (CAAIXSJLJJ-2021-016B, CAAIXSJLJJ-2022-001A), Anhui Province Key Research and Development Program (202104a05020007), USTC-IAT Application Sci. & Tech. Achievement Cultivation Program (JL06521001Y), Sci. & Tech. Innovation Special Zone (20-163-14-LZ-001-004-01). Bo Han was supported by the NSFC Young Scientists Fund No. 62006202, NSFC General Program No. 62376235, and Guangdong Basic and Applied Basic Research Foundation No. 2022A1515011652. Tongliang Liu was partially supported by the following Australian Research Council projects: FT220100318, DP220102121, LP220100527, LP220200949, and IC190100031.

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

# A  CIFAR20-SDN

## A.1  Construction Details

Subclass-dominant label noise (SDN) is designed to mimic human labeling errors, which often emerge from distinct characteristics of examples. To identify such examples, we employ CIFAR-100, which provides two types of labels: class labels and sub-class labels. Samples belonging to the same subclass are considered to possess distinct characteristics. During the generation of the SDN dataset, we randomly flip a specified number of the last subclass labels within each class to the subsequent class. For instance, the labels of the "whale" subclass within the "aquatic mammals" class are probabilistically flipped to the "fish" class. Table 6 illustrates the flipped sub-classes and the relationships between classes and sub-classes in CIFAR-100. For simplicity, we refer to this newly generated SDN dataset as CIFAR20-SDN.

Table 6: The table outlines the comprehensive structure of the classes within CIFAR20-SDN. Sub-classes marked in bold denote those from which examples are selectively flipped.

| Class name | Abbreviation | Sub-classes name |
|---|---|---|
| aquatic mammals | aquatic | beaver, dolphin, otter, seal, **whale** |
| fish | fish | aquarium fish, flatfish, ray, shark,**trout** |
| flowers | flowers | orchids, poppies, roses, sunflowers, **tulips** |
| food containers | containers | bottles, bowls, cans, cups, **plates** |
| fruit and vegetables | fruit | apples, mushrooms, oranges, pears, **sweet peppers** |
| household electrical devices | electrical | clock, computer keyboard, lamp, telephone, **television** |
| household furniture | furniture | bed, chair, couch, table, **wardrobe** |
| insects | insects | bee, beetle, butterfly, caterpillar, **cockroach** |
| large carnivores | carnivores | bear, leopard, lion, tiger, **wolf** |
| large man-made outdoor things | man-made | bridge, castle, house, road, **skyscraper** |
| large natural outdoor scenes | natural | cloud, forest, mountain, plain, **sea** |
| large omnivores and herbivores | omnivores | camel, cattle, chimpanzee, elephant, **kangaroo** |
| medium-sized mammals | m-mammals | fox, porcupine, possum, raccoon, **skunk** |
| non-insect invertebrates | invertebrates | crab, lobster, snail, spider, **worm** |
| people | people | baby, boy, girl, man, **woman** |
| reptiles | reptiles | crocodile, dinosaur, lizard, snake, **turtle** |
| small mammals | s-mammals | hamster, mouse, rabbit, shrew, **squirrel** |
| trees | trees | maple, oak, palm, pine, **willow** |
| vehicles 1 | vehicles 1 | bicycle, bus, motorcycle, pickup truck, **train** |
| vehicles 2 | vehicles 2 | lawn-mower, rocket, streetcar, tank, **tractor** |

## A.2  Diverse Challenges across Different Classes

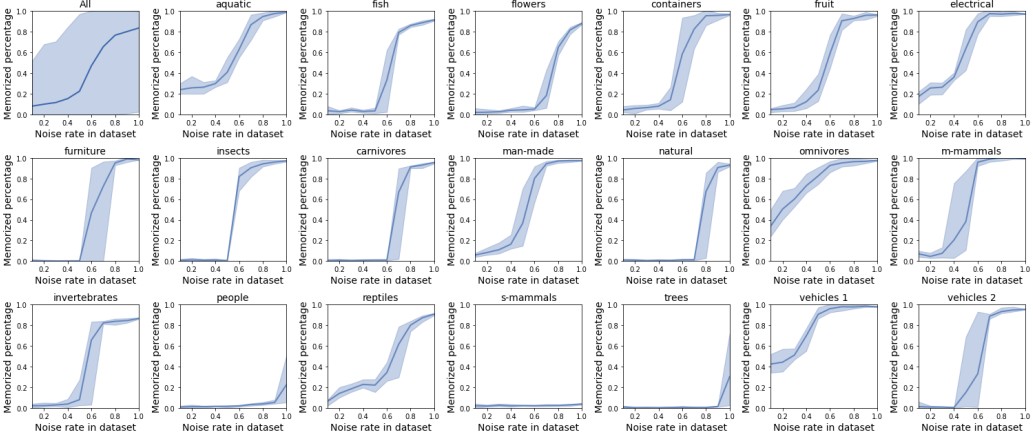

Figure 9: The figure shows the effectiveness of early stopping across different classes and various noise rates. When the noise rate in the dataset is small, early stopping performs well in all classes. However, when it is over 50%, memorized mislabeled samples increases rapidly.

CIFAR20-SDN presents a multifaceted landscape with its array of sub-classes, paving the way for a comprehensive study of SDN and a deeper insight into the intricacies of early stopping. In our experiments, we adjust the overall noise rate from 0% to 20%. This implies that within each specific subclass, the noise rate varies from 0% to 100%.

As illustrated in Figure 9, the effectiveness of early stopping in addressing SDN is notably inconsistent, heavily contingent on the specific subclass. Our analysis identifies two primary factors that significantly influence the success of early stopping. First, when the mislabeled subclass shares substantial semantic similarities with the clean class, distinguishing mislabeled examples from the clean class becomes challenging. For instance, mislabeling an instance from the "whale" subclass as a "fish" class is difficult to discern from other "fish" instances. Second, when classes have low diversity, the network tends to identify outliers, thus facilitating the detection of mislabeled examples, such as "trees" and "flowers". Conversely, distinguishing between correctly labeled and mislabeled examples becomes considerably more challenging in classes with higher diversity, such as "insects".

### A.3 Clusters Effects

We train a ResNet-18 on CIFAR20-SDN for 200 epochs with an overall noise rate of 18% (SDN-18). Afterward, we utilized t-SNE to visualize all 45,000 training examples. By comparing Figure 10 (a) and (b), we can pinpoint the noise locations, specifically evident in the small clusters. Notably, the vast majority of the mislabeled examples tend to cluster together and usually maintain a distance from their noisy class. This pattern suggests that clustering algorithms, such as DBSCAN, can effectively identify them. However, when the semantics of mislabeled examples closely match those of the noisy class, these mislabeled examples' clusters may not be distinct from the main cluster of the noisy class, which is a limitation of our proposed method.

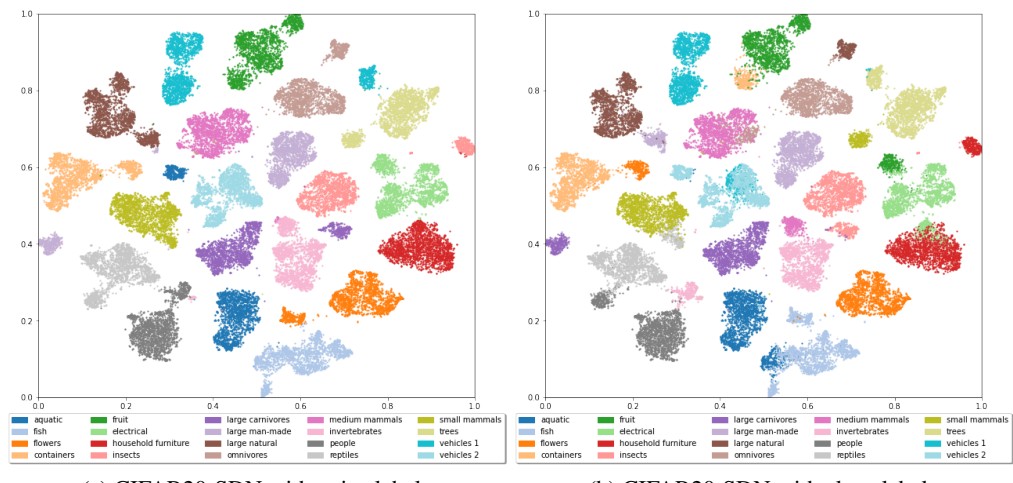

(a) CIFAR20-SDN with noisy labels          (b) CIFAR20-SDN with clean labels

Figure 10: The figure shows the distribution of training examples after 200 epochs on CIFAR20-SDN. We can find that the large clusters represent examples with correct labels, whereas the smaller clusters indicate those with incorrect labels.

## B  More Details about Experimental Settings

**Data preprocessing**. For all experiments, we employ simple data augmentation, including random crop and horizontal flip. For the Clothing-1M dataset, images are initially resized to a resolution of 256 x 256 pixels. Subsequently, a random crop is applied to produce images of 224 x 224 pixels, which are then followed by a random horizontal flip.

**Hyper-parameters of semi-supervised learning**. We follow [30, 3], and combine NoiseCluster with the semi-supervised learning method, MixMatch [5]. For experiments with SSL, we set $K = 2$, $T = 0.5$, and $\lambda_u = 0$. $\alpha$ begins at $4$ and drops to $0.75$ after half of the total number of epochs.

**Clothing1M**. Due to the large size of the Clothing1M dataset [52], which consists of one million training examples, the computational time for t-SNE is significantly extended. This volume of data also results in changes to the hyper-parameters. To address these issues, we split the Clothing1M training data into two parts: a larger subset containing 950,000 examples and a smaller one with 50,000 examples. We initially employ the larger subset for training a model with early stopping. Subsequently, based on the trained model, we select and correct confident data from the smaller subset. This corrected data is then used for the remainder of the training process. Note that the model used in the Clothing1M experiments has been pre-trained with the ImageNet-1k dataset [28], leading to the generation of separable representations. As a result, late stopping is omitted in this case. When applying NoiseCluster to a model trained from scratch, it is advisable to employ late stopping.

## C    Additional Experiments

### C.1    Ablation Study for Stopping Epoch

The effectiveness of NoiseCluster relies on long-trained representations derived from a model trained with later stopping, which stops the training process significantly later than what is common with early stopping. This raises two pertinent questions: How is the stopping epoch determined for later stopping? And does it display the same sensitivity as early stopping? To address these questions, we conduct a series of ablation experiments, adjusting the stopping epoch from 20th to 200th epochs.

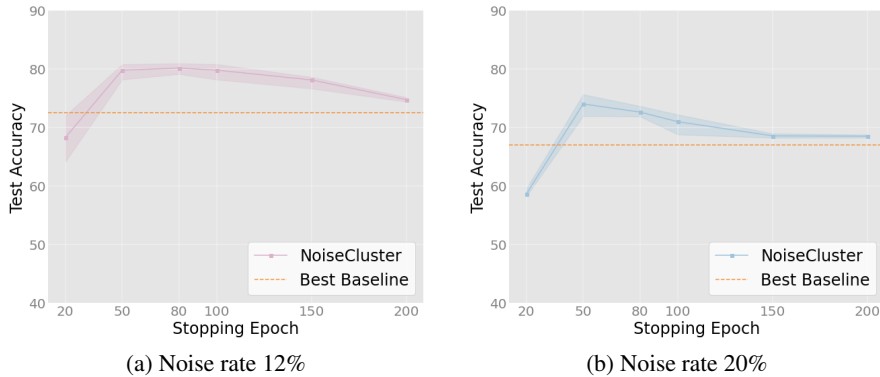

(a) Noise rate 12%                    (b) Noise rate 20%

Figure 11: Ablation analyses of stopping epoch. The mean and standard deviation (percentage) computed over five runs are presented.

As shown in Figure 11, NoiseCluster does not perform well with representations obtained at the 20th epoch, indicating that the effectiveness of NoiseCluster hinges on long-trained representations. Additionally, we can observe that NoiseCluster maintains a stable performance between the 50th and 100th epochs. This span is considerably wider compared to early stopping, which typically require a precise stopping epoch or operate within a relatively narrow range [30, 34, 3]. To delve deeper into the impact of the stopping epoch, we expand the training period for later stopping, and stop the training up to the 200th epoch. Despite a slight drop in the performance of NoiseCluster, attributable to the degradation of representation quality, it remains superior to the best baseline, CDR[49]. These experiments suggest that the choice of the stopping epoch in NoiseCluster can be made with considerable flexibility, displaying minimal sensitivity and negligible impact on the performance of NoiseCluster over a broad range.

### C.2    Explore Noise-modeling-based Approaches on SDN

In this section, we further investigate into why noise-modeling-based approaches are ineffective for SDN. Methods based on noise modeling build a transition matrix between the clean distribution and noisy distribution [40, 51]. We utilize the state-of-the-art method, VolMinNet [32], on CIFAR20-SDN with a noise rate 18%. Following the original paper [32], we set $\hat{T}$ to -2 and -4.5, respectively.

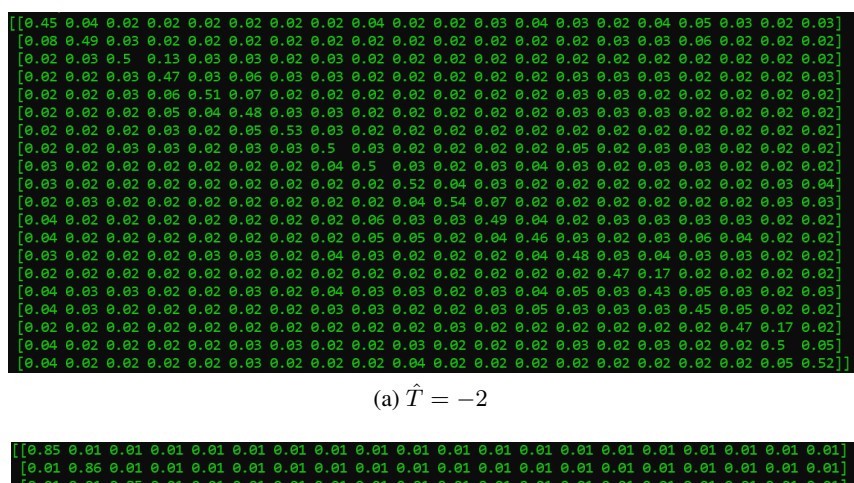

(a) $\hat{T} = -2$

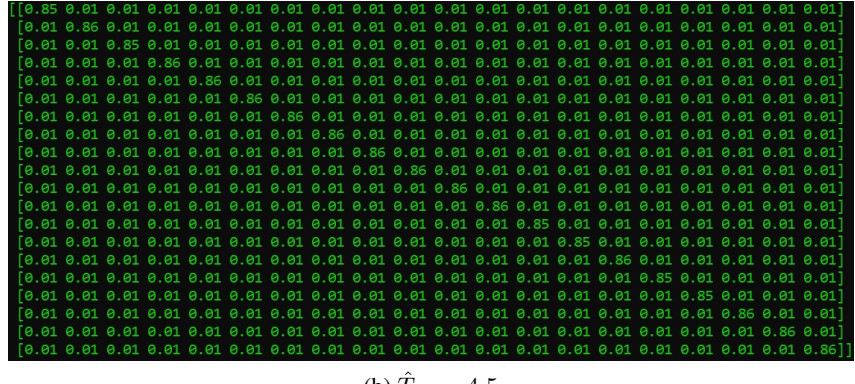

(b) $\hat{T} = -4.5$

Figure 12: **(a, b)** display the best estimated transition matrices during the training process with $\hat{T} = -2$ and $\hat{T} = -4.5$. The estimated noisy class-posterior probabilities are seen to be widespread across all classes, which is inconsistent with the generation of CIFAR20-SDN.

As depicted in Figure 12, it is evident that $\hat{T}$ holds the capability to modulate the extent of estimated noisy class-posterior probabilities within the transition matrix. For example, with $\hat{T} = -2$, VolMin-Net predicts higher noisy class-posterior probabilities in the transition matrix compared to that with $\hat{T} = -4.5$. However, neither of these $\hat{T}$ settings successfully enable an accurate estimation of the transition matrix. Consequently, these inaccurately estimated transition matrices result in inferior performance on SDN.

### C.3 Training Time Comparison

Table 7: Training time comparison on CIFAR20-SDN with SDN-12. All methods run on four core CPU and a single Nvidia V100.

| NOISECLUSTER | JOINTOPTIM | TOPOFILER | CDR |
|:---:|:---:|:---:|:---:|
| 1.7H | 2.5H | 1.5H | 5.5H |

| NOISECLUSTER+ | DIVIDEMIX | PES SEMI | F-DIVIDEMIX |
|:---:|:---:|:---:|:---:|
| 2H | 5.5H | 3.1H | 14.2H |

We evaluate the efficiency of our proposed method with state-of-the-art baselines. Table 7 displays the total training time on CIFAR20-SDN. We can see that the training time of NoiseCluster is comparable to that of TopoFiler [46] and faster than JointOptim [43] and CDR[49]. Furthermore, NoiseCluster+ exhibits substantial efficiency compared with other baselines that employ SSL. For instance, it requires less than half the time needed by DivideMix [30].

# D   Exploring SDN in Real-world Datasets

Due to space constraints in the main paper, we presented only a limited number of SDN examples. In this subsection, we provide additional visual evidence to further demonstrate the presence of SDN in the Clothing1M and WebVision datasets.

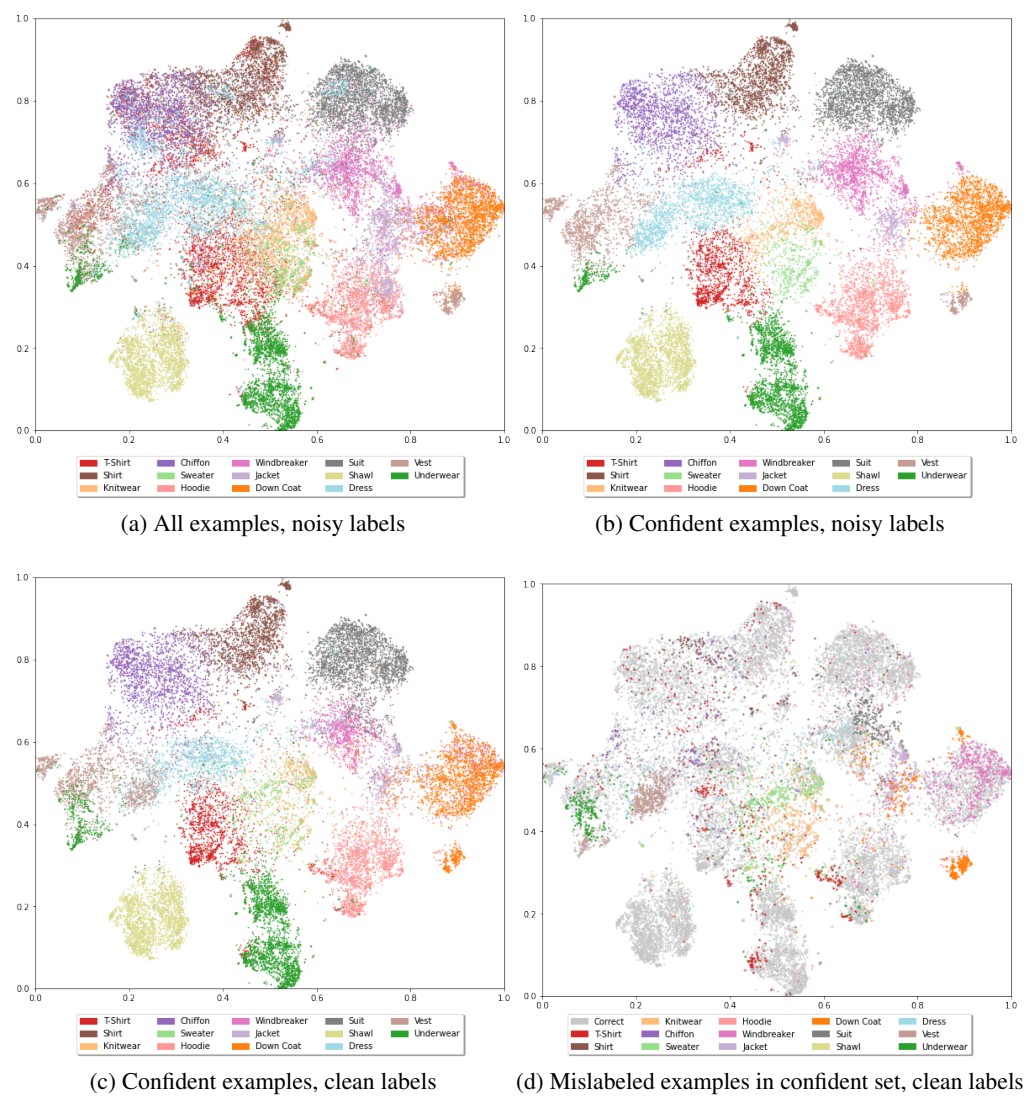

(a) All examples, noisy labels

(b) Confident examples, noisy labels

(c) Confident examples, clean labels

(d) Mislabeled examples in confident set, clean labels

Figure 13: (a) We visualize the features of all the examples in the evaluation set with noisy labels. (b) By removing unconfident examples, the features of confident examples are better separated. (c) We visualize the features of confident examples with clean labels. (d) We visualize the features of mislabeled examples in the confident set with clean labels. It is evident that the majority of mislabeled examples cluster together, confirming our observation of the clustering phenomenon in long-trained representations.

## D.1   SDN in Clothing1M

Figure 13 presents a step-by-step exploration process for the Clothing1M dataset, demonstrating how features of mislabeled examples are clustered together based on their semantics. Given the plethora of terms available for different clothing items, we can use accurate names to pinpoint these sub-classes. We highlight three sub-classes prominently impacted by label noise: "Sleep T-Shirt," "Leather Jacket," and "Down Vest," as shown in Figures 14, 15, and 16.

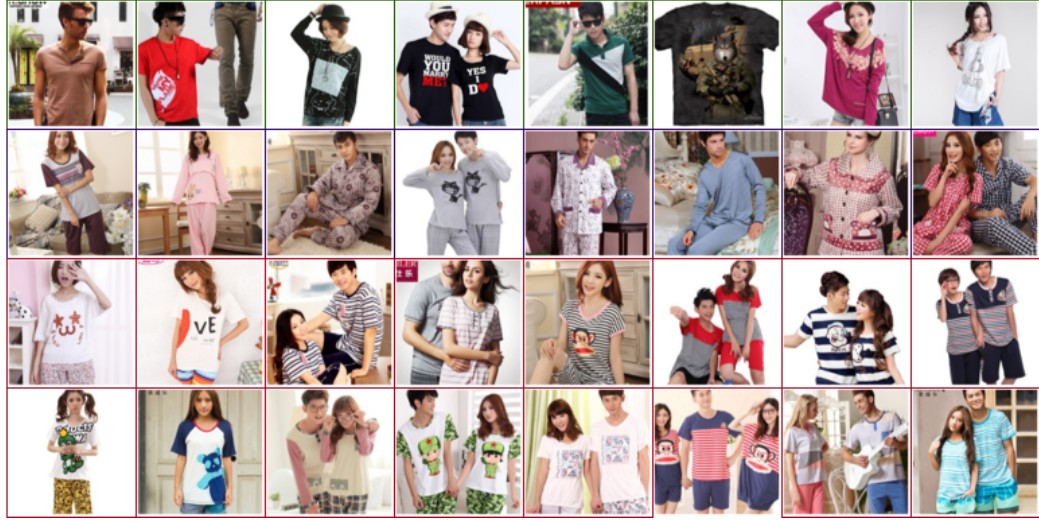

Figure 14: Subclass: Sleep T-Shirt, positioned at coordinates (X:0.45, Y:0.1). The first row displays images from the "T-Shirt" class. The second row showcases those from the "Underwear" class, while the third and fourth rows present images mislabeled as "Underwear". For the mislabeled pictures, it's evident that these images depict a type of shirt meant for home use, Sleep T-Shirt. After comparing images in the test set and Chinese label names, we put the correct names in parentheses.

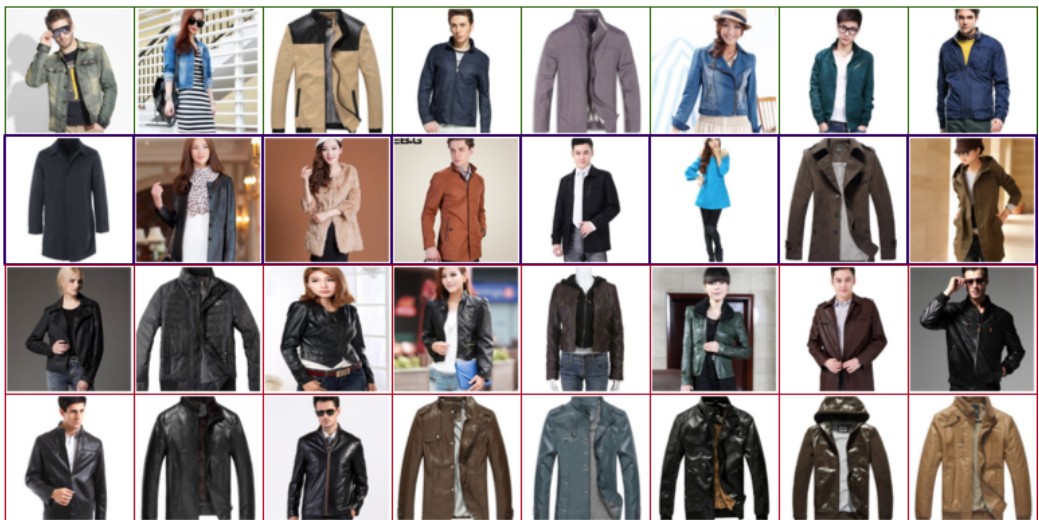

Figure 15: Subclass: Leather Jacket, positioned at coordinates (X:0.75, Y:0.6). The first row images are from the "Jacket" class, and the second row presents images correctly labeled as the "Windbreaker" class (correct name:"trench coat"). Clearly, these mislabeled images belong to the "Leather Jacket" subclass within the "Jacket" category.

We can also observe that some mislabeled examples are not clustered together, but are instead evenly dispersed in some classes. It is important to understand that in the real world, clothing categories are not solely distinguished by visual features. For instance, the features of "Sweater" and "Knitwear" are intertwined, with the main distinction being the material, not the visual appearance. Similarly, the categories "Underwear" and "Vest" (correct name:"singlet" or "tank top") lead to confusion since some garments can serve multiple purposes.

**Removing correctly labeled examples.** Early stopping not only helps remove mislabeled examples but also inadvertently excludes those that are correctly labeled when SDN exists. A closer inspection of instances within the "Leather Jacket" cluster confirms the issue: out of the total examples, 313 are mislabeled, with only 23 correctly labeled. This finding indicates that more than 90% of the "Leather

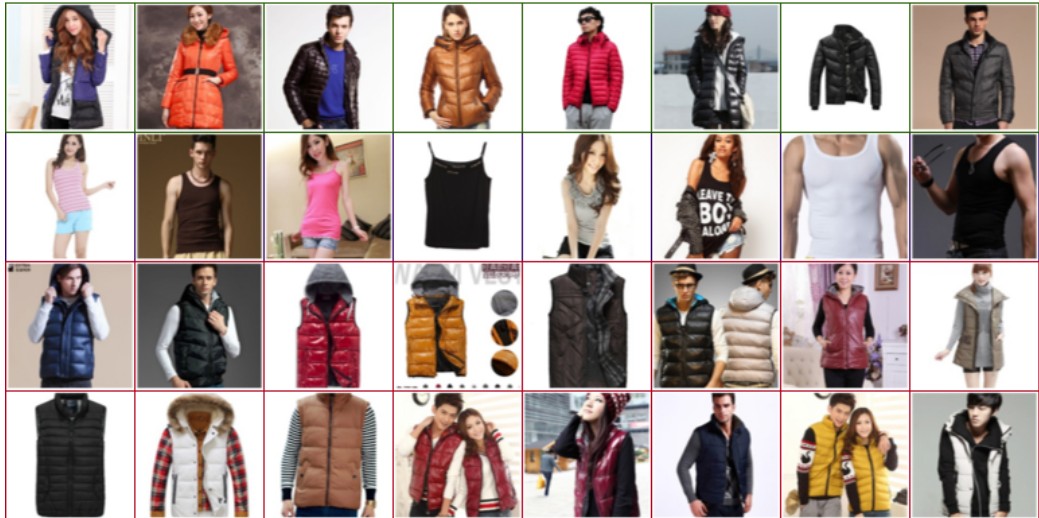

Figure 16: Subclass: Down Vest, positioned at coordinates (X:0.9, Y:0.3). The first row displays images from the "Down coat" class, while the second row showcases those from the "Vest" class (correct name:"singlet" or "tank top"). When the "Vest" class is absent, these mislabeled images should be classified as "Down Vest" within the "Down coat" class.

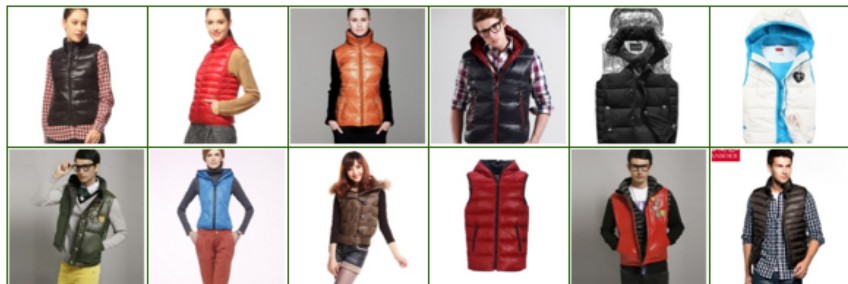

Figure 17: We discover 23 instances of "Leather Jacket" within the unconfident set. Despite being correctly labeled, these images are omitted from the confident set due to using early stopping. This process results in a purer cluster that predominantly contains mislabeled examples.

Jacket" examples are incorrectly labeled in the original dataset. Furthermore, the implementation of early stopping leads to the removal of the small fraction of correctly labeled examples from the confident set, as shown in Figure 17, pushing the noise rate from over 90% to close to 100%.

## D.2   SDN in WebVision

Table 8: Top-1 performance of our re-implemented DivideMix models alongside the pre-trained models, as assessed on the ImageNet and WebVision validation datasets. By employing an ensemble of pre-trained models, we generate pseudo clean labels to effectively identify mislabeled examples.

| Methods | Output Class | WebVision | ImageNet |
|---|---|---|---|
| DivideMix | 50 | 77.34 | 75.2 |
| DivideMix (reproduce) | 50 | 77.04 | 74.24 |
| Regnet_y_128gf (IMAGENET1K_SWAG_E2E_V1) | 1000 | 86.64 | 91.12 |
| ViT_H_14 (IMAGENET1K_SWAG_E2E_V1) | 1000 | 86.08 | 91.2 |
| Ensemble pretrained models | 1000 | 86.96 | 91.6 |

To extend our evaluation of SDN prevalence, we engage in experiments using another real-world dataset: WebVision V1.0 [31]. Following the setting presented in [8], we utilize the first 50 classes of WebVision, known as "mini WebVision". To demonstrate the flexibility of integrating with existing

early stopping methods, we employ DivideMix to select confident examples and extract long-trained representations from two networks, which have been trained for 80 epochs. The extracted features are presented in Figure 18. We utilize an ensemble of pre-trained models to produce pseudo clean labels, and their performance is depicted in Table 8.

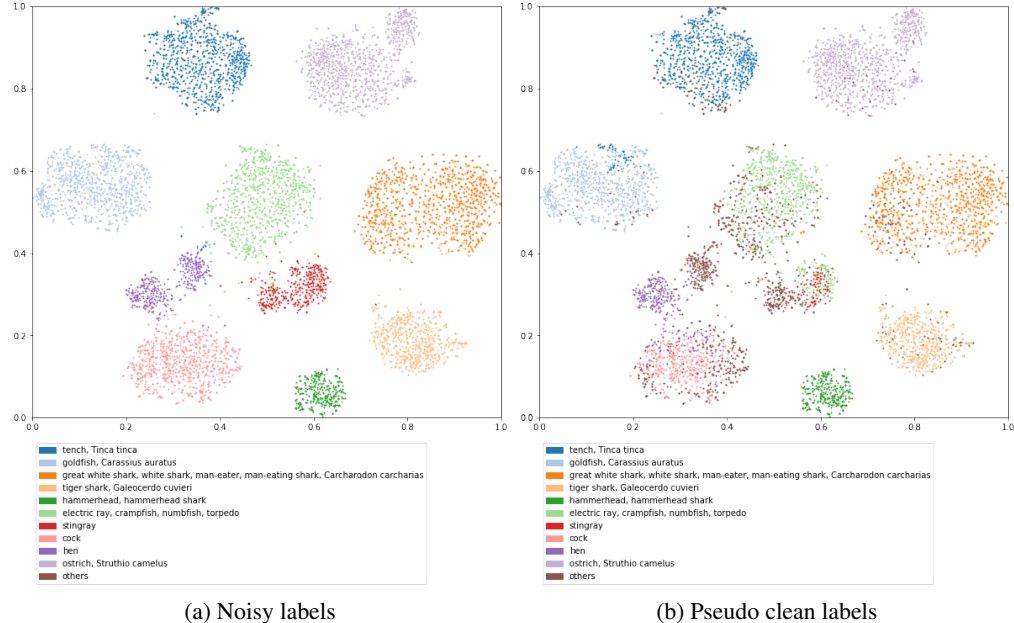

      (a) Noisy labels                               (b) Pseudo clean labels

Figure 18: Due to the absence of clean labels for the training data in WebVision, we employ two pre-trained models from PyTorch to generate pseudo-clean labels. While the pseudo-clean labels may not be accurate, they suffice to provide a general overview of the distribution of mislabeled examples. We then visualize confident features for the first ten classes using t-SNE, comparing noisy labels with the pseudo-clean ones. The category "other" represents examples where predictions from the pre-trained models are not in the first ten classes.

In Figures 19, 20, and 21, it is evident that, despite clear discrepancies between mislabeled and correctly labeled images, methods based on early stopping struggle to effectively identify these mislabeled examples. Moreover, these mislabeled samples do not fall into conventional categories such as "human" or "cars". Instead, they cluster according to different criteria, sharing similarities in attributes like shape and color with their correctly labeled counterparts. By considering out-of-domain examples as a single, unified class—termed "other", the mislabeled examples within each class can be interpreted as sub-classes of this "other" category.

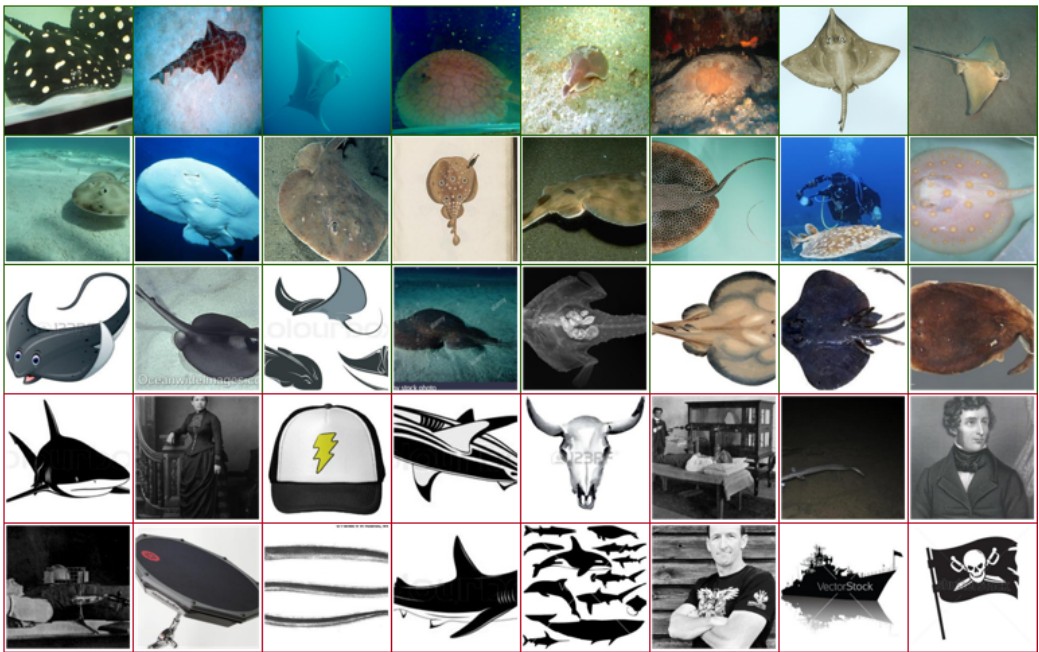

Figure 19: Explore images categorized under the "Electric Ray" class. The first row showcases images sourced from the ImageNet validation set, while the second row features images from the WebVision validation set. Correctly labeled training images are in the third row, whereas the fourth and fifth rows illustrate instances of mislabeling in the training set. We can find majority of mislabeled images are in "black and white" pattern, which are close to some examples in the third row, so we call it "black and white" subclass.

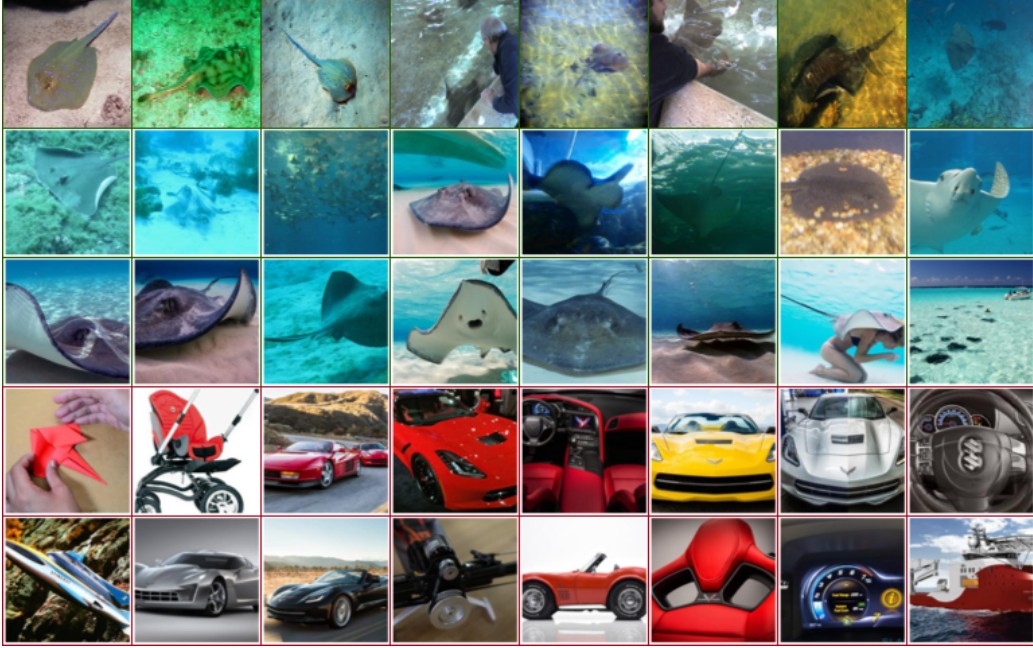

Figure 20: The row structure remains consistent with the previous format. Notably, numerous cars are incorrectly classified in this category, alongside the presence of wheels and ships. The leading attribute contributing to this misclassification is likely the presence of "curves".

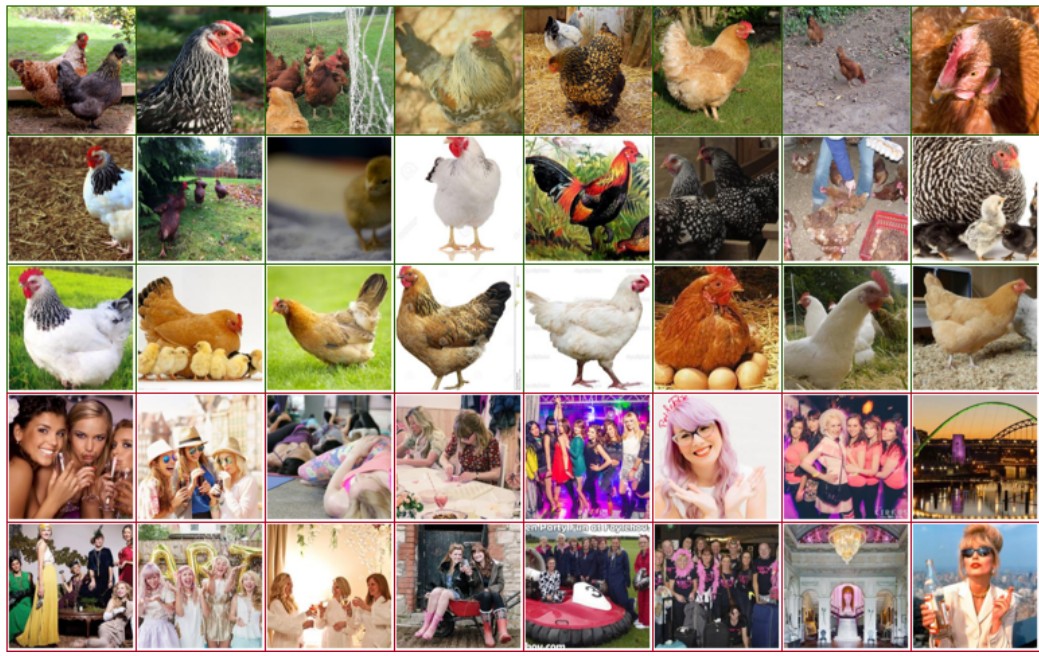

Figure 21: The row structure remains consistent with the previous format. The mislabeled subclass may derive from color attributes, such as pink or white, which serve as the decisive classification features for the "hen" category.

