# OpenReview forum: "Subclass-Dominant Label Noise: A Counterexample for the Success of Early Stopping"
_NeurIPS.cc/2023/Conference — NeurIPS 2023 poster_

### Official Review · Reviewer_tXVq · 2023-07-05

**Soundness:** 3 good
**Presentation:** 3 good
**Contribution:** 2 fair
**Rating:** 5
**Confidence:** 4

**Summary:**

This paper proposes a new type of noisy labels called subclass-dominant noisy labels and introduces an algorithm called NoiseCluster based on this noisy label modeling. In the experimental section, the authors demonstrate the superiority of NoiseCluster over previous approaches in the subclass-dominant noisy label cases. Additionally, NoiseCluster exhibits better performance than previous algorithms on the real-world Clothing1M dataset.

**Strengths:**

- Introduces a new type of noisy label modeling.
- Well-written paper.
- Conducts extensive experiments.


**Weaknesses:**

- The justification for the proposed subclass-dominant noisy label model is lacking. It is unclear whether this type of noisy labels frequently occurs in the real-world. For instance, it would be helpful to investigate if the Clothing1M dataset exhibits similar phenomena.
- The analysis should include the performance of NoiseCluster on other types of noisy label models such as symmetric, instance, or asymmetric. This would provide a more comprehensive evaluation of the algorithm.
- Further analysis on real-world datasets is necessary. While the authors evaluate NoiseCluster on the Clothing1M dataset, it would be beneficial to assess its performance on other real-world benchmarks such as WebVision, Food101N, and similar datasets.

**Questions:**

-

**Limitations:**

The limitations of this paper are summarized in the "Question" and "Weakness" sections

---

> ### Author Rebuttal · Authors · 2023-08-10
>
> >** **
> W1: The justification for the proposed subclass-dominant noisy label model is lacking ... investigate if the Clothing1M dataset exhibits similar phenomena.
>
> A1: Thank you for your insightful feedback. We highly agree with the justifying the presence of SDN and the noise clustering phenomena is pivotal for our study. For more existence of SDN, we response it in General Question 2. Detailed results are available in the attached one-page PDF.
>
> >** **
> W2: The analysis should include the performance of NoiseCluster on other types of noisy label models such as symmetric, instance, or asymmetric. This would provide a more comprehensive evaluation of the algorithm.
>
> A2: In this study, our primary focus is on examining a specific type of real-world label noise: Subclass Dominant label Noise (SDN). NoiseCluster has been specifically designed to tackle SDN.
>
> For other types of synthetic label noise, such as symmetric, instance,  or asymmetric, we response it in General Question 1.
>
> >** **
> W3: Further analysis on real-world datasets is necessary ... assess its performance on other real-world benchmarks such as WebVision, Food101N, and similar datasets.
>
> A3: We agree that evaluating our method, NoiseCluster, on additional real-world benchmarks would strengthen our findings and provide a more comprehensive understanding of SDN. Following your suggestion, we delve into SDN within the WebVision dataset. We find that SDN is prevalent across classes within the mini WebVision dataset. Moreover, clustering effects, specific to SDN, are distinctly evident therein. We pinpointed three unique SDN cases in WebVision and have provided visualizations of their associated images. These can be viewed in the attached one-page PDF.

---

> > ### Comment · Reviewer_tXVq · 2023-08-17
> > **Answer for official comment**
> >
> > Thank you for your detailed responses. I have read the responses to my question, and I will maintain the score regarding this. (rating 5)

---

> > > ### Author Response · Authors · 2023-08-19
> > >
> > > Thank you for your invalue suggestions. Due to the page limitions during rebuttal, we will provide more evidences for the existence of SDN and its negative imparts into the upcoming version.

---

### Official Review · Reviewer_Agy8 · 2023-07-05

**Soundness:** 3 good
**Presentation:** 3 good
**Contribution:** 2 fair
**Rating:** 6
**Confidence:** 3

**Summary:**

This paper investigates the impact of early stopping on models trained with noisy labels and introduces a new type of label noise called subclass-dominant label noise (SDN). The experiments reveal that later stopping during training can better capture the high-level semantics of noisy examples. Building upon this finding, the authors propose a novel approach named NoiseCluster, which utilizes the geometric structures of long-trained representations to detect and correct SDN. The experimental results demonstrate that NoiseCluster outperforms several label noise robust methods on both SDN and Clothing-1M datasets.

**Strengths:**

- This paper introduces a novel type of label noise called subclass-dominant label noise (SDN), providing a well-motivated and insightful perspective on the limitations of early stopping in the presence of SDN. By demonstrating the inapplicability of early stopping for training with this type of label noise, the authors shed light on the challenges faced by early stopping in handling real world label noise.

- Experimental results showcase the superior performance of the proposed method on both synthetic and real-world datasets, highlighting its effectiveness in addressing the issues posed by SDN.


**Weaknesses:**

- The paper would benefit from a comparison with instance-dependent label noise robust methods, which could provide a comprehensive evaluation of the proposed approach.
- The evaluation of methods in this study is limited to a single synthetic dataset and one real-world dataset. Assessing the proposed approach on a wider range of datasets can further demonstrate the generalizability and robustness of their method in various practical scenarios.

**Questions:**

 Could the authors provide further clarification on the differences between SDN and IDN? Moreover, given that there have been existing studies on IDN, what is the significance of specifically investigating and studying SDN?

**Limitations:**

please refer to the Weaknesses

---

> ### Author Rebuttal · Authors · 2023-08-10
>
> >** **
> Q1: The paper would benefit from a comparison with instance-dependent label noise robust methods, which could provide a comprehensive evaluation of the proposed approach.
>
> A1: Thank you for your insightful suggestion. In this work, we have included 12 baselines, many of which have been demonstrated to perform well on IDN. For example, PTD-R-V and BLTM-V are two methods specifically designed to address IDN, and both DivideMix and PES(semi) show strong performance on the CIFAR-N dataset, which contains human annotations. Despite their success in these settings, all of these baselines encounter serious difficulties when dealing with SDN, which validates the effectiveness of NoiseCluster in handling SDN.
>
> >** **
> Q2: Assessing the proposed approach on a wider range of datasets can further demonstrate the generalizability and robustness of their method in various practical scenarios.
>
> A2: We highly agree with the importance of evaluating our method on a broader range of real-world datasets. In pursuit of this, we undertook additional experiments on mini WebVision and revisited our experiments on Clothing1M to confirm the presence of SDN and the noise clustering phenomenon. Our observations indicate that SDN is widespread across classes in both datasets. Moreover, clustering effects inherent to SDN are clearly observable, affirming the efficacy of our proposed method in detecting SDN. We've pinpointed six unique SDN cases from WebVision and Clothing1M and closely examined their corresponding images. These findings are detailed in the attached one-page PDF.
>
> For other types of synthetic label noise, such as symmetric or asymmetric, we response it in General Question 1.
>
> >** **
> Q3: Could the authors provide further clarification on the differences between SDN and IDN? Moreover, given that there have been existing studies on IDN, what is the significance of specifically investigating and studying SDN?
>
> A3: We truly appreciate this insightful question. Since Instance-Dependent Noise (IDN) is still an area under active research, multiple explanations for IDN may exist. Below, we offer our perspective.
>
> IDN is derived from a noise modelling perspective and its definition is exceptionally flexible, encompassing all types of label noise, including SDN. This comprehensive approach enables the pursuit of a universal solution that can address all forms of label noise. However, this broad approach also presents challenges, such as defining a generation method that accurately mimics all real-world label noise types, or identifying typical real-world datasets that include all forms of label noise.
>
> On the other hand, SDN is a distinct type of label noise we have identified empirically. We came across specific mislabeled examples resistant to early stopping yet exhibiting recurring patterns. Based on these findings, we deduced the underlying causes, articulated the concept of SDN, and embarked on thorough experiments to explore its properties, all of which are elaborated upon in our paper. The value of our SDN study stems from its emulation of a prevalent type of real-world label noise, notably observed after early stopping.
>
> Overall, since current methods may struggle with SDN, we believe specifying SDN will aid in the study of IDN.

---

### Official Review · Reviewer_t1fi · 2023-07-07

**Soundness:** 2 fair
**Presentation:** 2 fair
**Contribution:** 2 fair
**Rating:** 5
**Confidence:** 3

**Summary:**

In this work, the authors present a new type of label noise: subclass-dominant label noise. The authors show that the model trained over time can better capture such label noise in the feature space, and based on this idea, a clustering then correcting pseudo labels algorithm, NoiseCluster, is designed to identify and correct SDN. The proposed method can be combined with current label noise and semi-supervised learning methods, and is validated on the authors' constructed dataset, cifar20-SDN, and another real-world dataset, Clothing 1M.

**Strengths:**

- The paper is well-organized. The Introduction and related work sections are clear and well introduces the studied problem.

- According to the proposed subclass-dominant label noise characteristics, the authors constructed a corresponding dataset and proposed a methodology addressing it. The proposed approach is heuristically reasonable and applicable.

**Weaknesses:**

- My main concern is the existence of proposed subclass-dominant label noise (SDN) in the real world scenarios.

- The proposed method works well on the constructed dataset, CIFAR20-SDN. However, the performance improvement on real-world dataset, Clothing 1M is marginal. This makes the existence of SDN questionable.

- Lack of some analysis of clusting effects and results in experiments. Please refer to the Questions.

**Questions:**

- It will be more clear to describe the class and sub-class relations briefly, or giving a concrete example of CIFAR20-SDN in the main paper.

- Clustering Result: More analysis of clusting effects and results could make the proposed SDn and method more convincing. For example, on the CIFAR20-SDN dataset, does the clustering result consist with the manually created SDN noise? And what about the real-world dataset Clothing 1M?

**Limitations:**

The authors have mentioned their limitations in the conclusion section.

---

> ### Author Rebuttal · Authors · 2023-08-10
>
> >** **
> Q1: My main concern is the existence of proposed subclass-dominant label noise (SDN) in the real world scenarios.}
>
> A1: Your concern is important. The existence of SDN in real-world scenarios is fundamental to the study of SDN. Since this question is so important, we reponse it in General Question 2.
>
> After the publication of this paper, we plan to create a website to showcase the collections of SDN we've identified. The site will also offer an upload feature, allowing other users to share SDN instances they've found. This initiative aims to address this issue in real-world scenarios. Additionally,  on the website, we will acknowledge all the reviewers for their invaluable suggestions.
>
> >** **
> Q2: The proposed method works well on the constructed dataset, CIFAR20-SDN. However, the performance improvement on real-world dataset, Clothing1M is marginal. This makes the existence of SDN questionable.
>
> A2: We concede that the improvements on Clothing1M are not as much as those on CIFAR20-SDN, given the complexity of Clothing1M. However, when compared with SOTA methods such as ELR+ and DivideMix, which both leverage dual networks, MixUp, and semi-supervised techniques (techniques known to enhance performance in clean settings). The improvements made by NoiseCluster are remarkable, considering it doesn't rely on these supplemental techniques. Furthermore, a 0.7\% rise is also considerable, constituting over 10\% of the total 6.3\% improvement in comparison to the CE result. Given the straightforward approach of NoiseCluster, it's evident that this enhancement is purely due to its adept handling of SDN.
>
>
> >** **
> Q3: It will be more clear to describe the class and sub-class relations briefly, or giving a concrete example of CIFAR20-SDN in the main paper.
>
>
> A3: We agree that providing a concrete example of class and sub-class relations, particularly in the context of the CIFAR20-SDN dataset, would help in understanding the concept of Subclass-Dominant Label Noise (SDN). Here's a potential addition to the manuscript:
>
> The first category in CIFAR20 is termed "aquatic mammals," which is subdivided into five sub-classes: "beaver", "dolphin", "otter", "seal", and "whale". To introduce SDN, we randomly flip a significant proportion of labels within the "whale" subclass to the subsequent category, "fish". Subsequently, "whales" labeled as "fish" are considered as SDN.
>
> We hope this example makes the concept of class and subclass relations, as well as the Subclass-Dominant Label Noise, clearer. We will incorporate this explanation into the revised manuscript to facilitate better understanding for readers.
>
> >** **
> Q4: Clustering Result: More analysis of clusting effects and results could make the proposed SDn and method more convincing. For example, on the CIFAR20-SDN dataset, does the clustering result consist with the manually created SDN noise? And what about the real-world dataset Clothing 1M?
>
> A4: Thank you for your insightful suggestions. We agree that a more thorough analysis of the clustering effects and results could strengthen our claims and make our proposed SDN and method more convincing.
>
> To this end, we have added t-SNE images from CIFAR20-SDN, Clothing1M, and WebVision datasets in the attached one-page PDF. A comparison of t-SNE images from these datasets with the manually created CIFAR-10 SDN in Figure 1 in the main paper clearly demonstrates a consistent clustering phenomenon across all four datasets.

---

> > ### Comment · Reviewer_t1fi · 2023-08-16
> > **Thank you for the detailed response**
> >
> > I appreciate the author's detailed response. The additional figures and explanation of the datasets could largely justify the existence of SDN in some real-world datasets, which addresses my main concerns to some extent. I've also read the other reviewers' comments and the authors' rebuttal and decided to raise my score from 4 to 5.

---

> > > ### Author Response · Authors · 2023-08-17
> > >
> > > We deeply appreciate the time and effort you've dedicated to providing us with valuable insights and feedback regarding the presence of SDN in real-world datasets. Such contributions significantly bolster the integrity and solidity of our work. With gratitude, we are committed to integrating these invaluable suggestions into the upcoming version.

---

> ### Author Response · Authors · 2023-08-15
>
> We sincerely appreciate your invaluable insights into the existence of the proposed SDN in real-world scenarios, as well as the feedback on consistency experiments using real-world datasets and the manually created SDN dataset. We have conducted additional experiments and put the results in the one-page PDF. If there are any further areas of our research that need clarification or if there are additional queries you might have, we are more than willing to provide further explanations.

---

### Official Review · Reviewer_7AfS · 2023-07-07

**Soundness:** 2 fair
**Presentation:** 2 fair
**Contribution:** 3 good
**Rating:** 5
**Confidence:** 4

**Summary:**

In the paper, the authors uncover the phenomenon that mislabeled examples are quickly learned during the initial stages of training when Subclass-Dominant Label Noise (SDN) is present. This behavior hinders the effectiveness of early stopping-based robust methods. To address this issue, the authors propose an approach that does not rely on early stopping. The method involves identifying mislabeled examples by clustering the penultimate layer features and correcting them by assigning them labels of the closest class. Experiments are conducted to demonstrate the superiority of their method compared to existing approaches.

**Strengths:**

1.	The finding of the failure of early stopping under SDN is intriguing with practical implications, as SDN can be a common occurrence in practice.
2.	The proposed method has significant improvements over existing methods under SDN.

**Weaknesses:**

Overall, while I appreciate the first part of the paper that discusses the failure of early stopping under SDN, I have some concerns and suggestions regarding the other parts:
1. It would be beneficial to delve deeper into the phenomenon of early stopping’s failure under SDN, e.g., exploring why subclass dominance leads to wrong labels being learned quickly.
2. The advantage of long-trained representations may not be surprising, as training progress can naturally lead to more distinguishable clusters. It seems that this property is not specific to SDN.
3. Regarding the combination with SSL, it is unclear why there would be features that are not assigned any group index. Are these features the same as $U^c_{k-1}$ mentioned in section 4.1?
4. The Noisecluster+ method, where labels are not corrected but instead discarded (as those examples are treated as unlabeled), achieves better performance than the class-correction based method. Does this mean label correction is not as effective?
5. I am curious about the percentage of examples whose labels were successfully corrected by the label-correction method.
6. In Table 5, is the method without label correction simply vanilla training? If that is the case, vanilla training already accomplishes the majority of the performance, and the performance improvements attributed to label correction appear relatively modest (1%). So the proposed method does not contribute much.
7. ELR should also be included in Table 2, as it does not require SSL (only ELR+ does).

**Questions:**

In addition to the questions raised in Weaknesses, I have the following additional questions:
1. Regarding Cloth1M, why was only one epoch used for training? Furthermore, what is the rationale behind training on 95% of the data first and then applying SDN with only 5%? Currently it is unclear why these modifications are made.
2. The method doesn't appear to be specifically designed for SDN alone, as identifying mislabeled examples and the label correction part make sense for general noise as well. How does the proposed method perform under normal noise?
3. Since the authors mentioned incorporating self-supervised learning, it would be valuable to discuss self-supervised pretraining, which has been shown to enhance robust methods [1][2][3][4]. As features are initially learned without labels, starting from a pretrained model might lower the chance of mislabeled examples in SDN being learned early. Therefore, I wonder if early stopping could potentially be effective with this pretraining even under SDN.

[1] Hendrycks, Dan, et al. "Using self-supervised learning can improve model robustness and uncertainty." Advances in neural information processing systems 32 (2019).

[2] Zheltonozhskii, Evgenii, et al. "Contrast to divide: Self-supervised pre-training for learning with noisy labels." Proceedings of the IEEE/CVF Winter Conference on Applications of Computer Vision. 2022.

[3] Xue, Yihao, Kyle Whitecross, and Baharan Mirzasoleiman. "Investigating why contrastive learning benefits robustness against label noise." International Conference on Machine Learning. PMLR, 2022.

[4] Ghosh, Aritra, and Andrew Lan. "Contrastive learning improves model robustness under label noise." Proceedings of the IEEE/CVF Conference on Computer Vision and Pattern Recognition. 2021.

**Limitations:**

The authors have discussed limitations in the paper.

---

> ### Author Rebuttal · Authors · 2023-08-10
>
> >** **
> Q1: It would be beneficial to delve deeper ... dominance leads to wrong labels being learned quickly.
>
> A1: Thanks for the insightful question. To explain why SDN is learned quickly, we must recall why a machine learning model can learn a generalized function and why the early stopping phenomenon exists in learning with noisy labels. In real life, ``class" denotes a group of objects with similar characteristics, defined by a set of rules. Deep networks, designed to identify these characteristics, facilitate the classification of real-world classes. In supervised learning, the classification task involves learning a function from prior labeled samples that can accurately predict unobserved samples, in accordance with the characteristics.
>
> This viewpoint sheds light on the early stopping phenomenon. A network can quickly learn a generalized function to recognize correctly labeled samples within a noisy dataset, as both observed and unobserved examples with clean labels share the same characteristics. On the other hand, the network encounters difficulty in finding a generalized function for mislabeled samples, especially if their generation method (e.g., random selection) fails to align with any real-world rule. As a result, the network is forced to memorize mislabeled samples individually to distinguish them from correctly labeled ones. This leads to the earlier learning of correctly labeled samples.
>
> However, real life provides various ways to organize objects, such as by status, color, shape, etc. For instance, in the experiments shown in Figure 1, examples within the airplane class can be divided into flying or landed categories based on their status. We flip the labels of landed airplanes to automobiles. The network can then quickly learn a generalized function to distinguish landed airplanes from flying ones by recognizing special features in landed airplanes, such as wheels or land. Overall, this may explain why SDN is learned quickly.
>
> Additionally, we have visualized SDN examples from both Clothing1M and WebVision datasets and included them in the attached one-page PDF. We believe these images will provide more hints on why SDN can be learned rapidly.
>
>
> >** **
> Q2: The advantage of long-trained representations may not be surprising, as training progress can naturally lead to more distinguishable clusters. It seems that this property is not specific to SDN.
>
> A2: (1) Thank you for your observation. In the self-supervised learning community, it has been widely recognized that long-trained representations become more distinguishable in alignment with their inherent features. In contrast, supervised learning typically sees representations of examples forming more distinct clusters based on their labels. To the best of my knowledge, no existing paper in the field of learning with noisy labels has discussed that long-trained representations will gravitate towards their inherent features (e.g., images) rather than conforming to their (noisy) labels, when using supervised learning.
>
> (2) Yes, this property is not unique to SDN. We have demonstrated that symmetric label noise exhibits a similar phenomenon, as shown in Figure 3(d). However, the representations resulting from symmetric label noise are significantly more complex and cannot be effectively clustered in a 2-dimensional space. Research methods, including TopoFilter and ELR, have highlighted the vital importance of the early learning phase in preventing the degradation of representations trained over extended periods. Thus, long-trained representations may not be suitable for all types of label noise.
>
>
> >** **
> Q3: Regarding the combination with SSL, it is unclear why there would be features that are not assigned any group index. Are these features the same as mentioned in section 4.1?
>
> A3: Thank you for bringing up this point. There are some peripheral points in t-SNE images, which are too sparse to constitute a new cluster and are distant from established groups. Then, these points of the features are not in any group, and are regarded as unlabeled examples for SSL. For a more comprehensive understanding of DBSCAN Noise, we recommend referring to the original DBSCAN paper.
>
>
> >** **
> Q4: The Noisecluster+ method, where labels are not corrected ... label correction is not as effective?
>
> A4: There is a potential confusion. Label correction is used in both NoiseCluster and NoiseCluster+. The term "NoiseCluster+" denotes NoiseCluster plus MixMatch.
>
>
> >** **
> Q5: I am curious about the percentage of examples whose labels were successfully corrected by the label-correction method.
>
> A5: The mean and standard deviation computed over five runs are presented in the below table.
>
> |Method| SDN-12 | SDN-16 | SDN-18 | SDN-20 |
> |:-----|:----:|:----:|:----:|:----:|
> |confident examples                  | 41536 (361) | 41484 (455)  | 41812 (501)  | 42245 (129) |
> |unconfident examples (DBSCAN Noise) | 3464 (361)  | 3516 (455)   | 3188 (501)   | 2755 (129)  |
> |total corrected examples            | 4774 (761)  | 4914 (544)   | 5958 (562)   | 6112 (609)  |
> |successful corrected examples       | 4153 (446)  | 4422 (763)   | 5040 (800)   | 3272 (762)  |
> |successful correct rate (\%)        | 87.50 (4.47) | 89.52 (5.65) | 84.25 (6.16) | 53.17 (8.47) |
>
> >** **
> Responses to questions Q6-Q10 can be found in General Questions due to character limitations.

---

> ### Author Response · Authors · 2023-08-16
>
> We sincerely appreciate the time and effort you've invested in thoroughly reviewing our paper, providing us with profound insights and constructive feedback. In response to the inquiries, we have addressed the following points:
>
> 1. Offered an in-depth explanation regarding the phenomenon of the early stopping's failure.
> 2. Elucidated the significance of long-trained representations discovered in label noise.
> 3. Carried out additional experiments to measure the successful correct rate and provided clarity on the techniques used for label correction, as used in both NoiseCluster and NoiseCluster+.
> 4. Delineated the reasons for specifical designing NoiseCluster to SDN.
> 5. Introduced a discussion on the challenges of SDN in Self-supervised learning.
>
> We also wish to emphasize the main contributions of this paper:
> 1. We first found a new kind of prevalent real-world label noise and created a dataset, CIFAR20-SDN, to successfully mimic its behavior (shown in the one-page PDF).
> 2. We first explicitly demonstrate the limitations of early stopping, which is widely used in SOTA methods.
> 3. We propose an innovative yet straightforward approach to tackle SDN. Furthermore, by integrating existing early stopping-based methods, NoiseCluster successfully detect and correct SDN in large, real-world noisy datasets.
>
> We will be integrating these modifications into the upcoming version. As the discussion period nears its end, we wish to make certain that all facets of our research are clear and well-understood. Are there any other areas within our study that you believe need further detail or clarification?

---

> > ### Comment · Reviewer_7AfS · 2023-08-21
> > **Thanks for your response**
> >
> > Thank you for clarifying, and I apologize for my late reply. The responses from the authors have addressed most of my concerns. As a result, I've raised my score.

---

### Author Rebuttal · Authors · 2023-08-10

>** **
General Questions

>** **
Q1: What is NoiseCluster's performance across different noisy label models, like symmetric, instance, and asymmetric?

A1: NoiseCluster has been specifically designed to tackle SDN, utilizing our proposed late stopping strategy. As evidenced in Table 1, representations under symmetric, instance, or asymmetric noise largely degrade over extended training periods. In such cases, early stopping is the typical strategy to prevent this degradation. Therefore, NoiseCluster might not match the performance of other SOTA methods for these specific noise types.

For a comprehensive algorithm to manage various label noises, we've proposed a combined strategy in Line 212. This involves pairing early stopping-based methods with NoiseCluster. Initially, the early stopping method tackles label noise such as symmetric, instance, or asymmetric. Subsequently, NoiseCluster continues to address SDN. By adopting this integrated approach, we ensure competitive performance against symmetric, instance, or asymmetric label noise, while also tackling a new type of real-world label noise.

>** **
Q2: Is the proposed subclass-dominant label noise (SDN) prevalent in real-world scenarios?

A2: We first address this question through logical reasoning, followed by showcasing examples of SDN from real-world datasets.

In real-world situations, label noise is not uniformly random (e.g., symmetric noise) but tends to follow certain patterns. For instance, samples from specific sub-classes may frequently be mislabeled, especially if annotators are unfamiliar with them. When humans annotate ambiguous samples, our approach isn't to assign arbitrary labels. Rather, we label based on our existing understanding. Hence, if an example is mistakenly labelled, other closely related examples (in the same subclass) could be similarly mislabeled. Labels generated from models may lead to this issue more serious, misclassifying similar samples in a consistent manner. This consistent mislabeling within sub-classes results in the prevalence of SDN in real-world datasets.

To confirm the presence of SDN in real-world datasets, we carry out experiments on Clothing1M and WebVision by carefully examining the images in clusters. Our observations indicate that SDN is widespread across classes in both datasets. These findings are showed in the attached one-page PDF.

>** **
More response to Reviewer 7AfS Question 6-10

>** **
Q6: In Table 5, is the method without label correction simply vanilla training?

A6: The result of 'Ours W/O CORRECTION' is not equivalent to vanilla training, which is denoted as 'CE' in Table 2. Our method consists of two crucial components: identification and label correction. When we say 'Ours W/O CORRECTION,' we mean that we do not correct the noisy labels, but we still identify potentially mislabeled examples. Once identified, we can exclude them from the confident examples. This removal is a simple yet effective strategy that results in a 6-8\% improvement. It is noted that the improvement of removal is built on the success of identifying SDN.

However, label correction remains a risky action, which may not work in a more complex situation. We will provide further clarification in Table 5 and Section 5.3. Thank you for pointing this out.

>** **
Q7: ELR should also be included in Table 2, as it does not require SSL (only ELR+ does).

A7: It's worth noting that ELR still retains some semi-supervised techniques, such as temporal ensembling and target probabilities. Since the performance of ELR+ falls below that of NoiseCluster (without semi-supervised techniques), the results might not differ significantly.

>** **
Q8: Regarding Cloth1M, why was only one epoch used for training? ... why these modifications are made.

A8: (1) Clothing1M is comprised of one million images with noisy labels. Even a single training epoch provides enough data for fine-turning a pretrained model, equivalent to 20 epochs on CIFAR-10. As a result, we only train the network for one epoch, making NoiseCluster not only the state-of-the-art method but also the fastest method on Clothing1M.

(2) Clothing1M is a real world noisy dataset, containing complex types of label noise, not only SDN. So, we employ a simple early stopping method, training on 95\% of the data, to remove other types of label noise, then apply NoiseCluster on the filtered dataset to continue to identify and correct SDN.

(3) The primary rationale for applying NoiseCluster to 5\% of the training data is to maintain the number of training examples on a par with CIFAR20-SDN, thereby minimizing the need for tuning DBSCAN hyperparameters. An additional consideration is the computational cost: the calculation of t-SNE for one million features takes around one hour, whereas it takes only about five minutes for 50k features.

>** **
Q9: The method doesn't appear to be specifically designed for SDN alone, ... under normal noise?

A9: NoiseCluster has been specifically designed to tackle SDN. For the performance of other types of synthetic label noise, such as symmetric or asymmetric, we response it in General Question 1.

>** **
Q10: it would be valuable to discuss self-supervised pretraining, ... even under SDN.

A10: It should be noted that our findings do not conflict with previous works. When using self-supervised pretrained models, the challenge of early stopping may be intensified. Since self-supervised learning is capable of uncovering underlying characteristics, it might enlarge the feature distances between sub-classes. This could lead to an acceleration in the speed of overfitting to SDN, making early stopping more ineffective. Supporting evidence can be drawn from the performance metrics of C2D (ELR+ with SimCLR) [2] on the Clothing1M dataset. C2D lags behind NoiseCluster by 0.93\% and even trails ELR+ by 0.23\%, despite the fact that the initialization of self-supervised pretrained models surpasses supervised learning.

---

### Decision · Program_Chairs · 2023-09-21

**Decision:**

Accept (poster)

**Comment:**

The paper studies subclass-dominant label noise, a type of noise where mislabeled examples dominate at least one class. The authors study the evolution of representations generated by long trained network, and show that similar examples can be clustered together. Based on this observation, they suggest a new algorithm for combating subclass-dominant label noise, by clustering examples together and then eliminating noisy examples based on their distance to other clustered examples.

It seems that, after discussion with the authors, all the reviewers are supporting the paper for acceptance. While some of the reviewers were concerned about the applicability of the noise model studied in the paper to real-world datasets, to my understanding the thorough response of the authors convinced the reviewers that this study is a worthy contribution.

In this situation, I recommend that the paper is accepted, assuming that the authors make the proper changes to the paper suggested in the reviews.